# $C^2M^3$: Cycle-Consistent Multi-Model Merging

**Donato Crisostomi**
Sapienza University of Rome
crisostomi@di.uniroma1.it

**Marco Fumero**
Institute of Science and Technology Austria
fumero@di.uniroma1.it

**Daniele Baieri**
Sapienza University of Rome
baieri@di.uniroma1.it

**Florian Bernard**
University of Bonn
fb@uni-bonn.de

**Emanuele Rodolà**
Sapienza University of Rome
rodola@di.uniroma1.it

## Abstract

In this paper, we present a novel data-free method for merging neural networks in weight space. Differently from most existing works, our method optimizes for the permutations of network neurons globally across all layers. This allows us to enforce cycle consistency of the permutations when merging $n \geq 3$ models, allowing circular compositions of permutations to be computed without accumulating error along the path. We qualitatively and quantitatively motivate the need for such a constraint, showing its benefits when merging sets of models in scenarios spanning varying architectures and datasets. We finally show that, when coupled with activation renormalization, our approach yields the best results in the task.

## 1 Introduction

In the early days of deep learning, modes — parameters corresponding to local minima of the loss landscape — were considered to be isolated. Being depicted as points at the bottom of convex valleys, they were thought to be separated by high-energy barriers that made the transition between them impossible. However, a series of recent works have gradually challenged this perspective, first showing that modes can be actually connected by paths of low energy [10, 14], and later that, in some cases, these paths may even be linear [13]. While linear paths in [13] could only be obtained after training the equally-initialized models for a few epochs, follow-up work [11] speculated that the isolation of modes is a result of the permutation symmetries of the neurons. In fact, given a layer $W_\ell$ of a fixed network $A$, a large number of functionally-equivalent networks can be obtained by permuting the neurons of $W_\ell$ by some permutation $P$ and then anti-permuting the columns of the subsequent layer $W_{\ell+1}$. This intuition led to the conjecture that all modes lie in the same convex region of the parameter space, denoted as *basin*, when taking into account all possible permutations of the neurons of a network.This motivated a series of works trying to align different modes by optimizing for the neuron permutations [1, 21, 29, 36]. This has strong implications for model merging, where different models, possibly trained with different initializations [1, 29, 34] or on different datasets and tasks [1, 36], are aggregated into a single one. In this work, we focus on the *data-free* setting, aligning networks based on some similarity function that is computed directly over the neurons themselves. To this end, we follow Ainsworth et al. [1] and formalize the problem of model merging as an assignment problem, proposing a new algorithm that is competitive with previous approaches while allowing global constraints to be enforced.

**The problem**   We investigate the problem of merging $n > 2$ models, noting that existing pairwise approaches such as [1] do not guarantee cycle consistency of the permutations (see Figure 1). As shown in Figure 2b and Figure 2a, going from a model $A$ to a model $C$ through a model $B$, and then mapping back to $A$, results in a different model than the starting one — specifically, the target

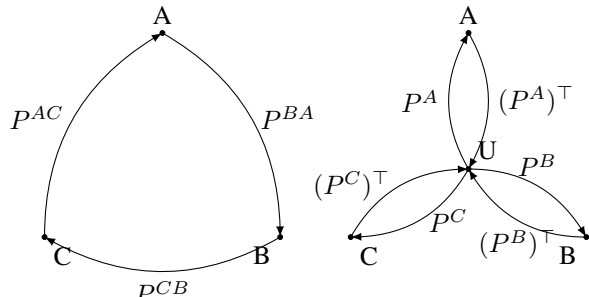

Figure 1: Cycle-Consistent Multi-Model Merging over three models $A, B, C$. **Left:** existing methods seek pairwise permutations that map between models; note that $P^{AC} \circ P^{CB} \circ P^{BA} \neq I$ in general, unless this is explicitly enforced. **Right:** our method computes permutations $P^A, P^B, P^C$ from each model to a *universe* $U$, such that a pairwise permutation $P^{BA}$ mapping $A$ to $B$ can be obtained as $P^{BA} = P^B(P^A)^\top$. This way, cycle-consistency is enforced by design and $P^{AC} \circ P^{CB} \circ P^{BA} = I$.

model ends up in a completely different basin. More formally, for these methods, the composition of permutations along any cycle does *not* result in the identity map. This also holds for the $n = 2$ case, where the permutations optimized to align model $A$ to model $B$ are not guaranteed to be the inverse of those mapping $B$ to $A$; this makes the alignment pipeline brittle, as it depends on an arbitrary choice of a mapping direction.

**Contribution** To address this issue, we introduce a novel alignment algorithm that works for the general case with $n \geq 2$ models, while *guaranteeing* cycle consistency. The key idea is to factorize each permutation mapping $B$ to $A$ as $P^{AB} = P^A(P^B)^\top$, where $(P^B)^\top$ maps $B$ to a common space denoted as *universe*, and $P^A$ maps from the universe back to $A$. This formulation ensures cycle consistency by design, as any cyclic composition of such permutations equals the identity.

Our numerical implementation is based on the Frank-Wolfe algorithm [12], and optimizes for the permutations of *all* the layers simultaneously at each step, naturally taking into account the inter-layer dependencies in the process. This desirable property is in contrast with other approaches such as Ainsworth et al. [1], which seek the optimal permutations for each layer separately, and thus can not ensure coherence across the entire network.

We run an extensive comparison of our approach with existing ones both in the standard pairwise setting and in merging $n > 2$ models, spanning a broad set of architectures and datasets. We then quantitatively measure the influence of architectural width, confirming the existing empirical evidence on its role in linear mode connectivity. Further, we assess how the performance of the merged model depends on the number of models to aggregate, and show that the decay is graceful. We finally analyze the basins defined by the models when mapped onto the universe, and investigate when and to what extent these are linearly connected.

Wrapping up, our contributions are four-fold:

- We propose a new data-free weight matching algorithm based on the Frank-Wolfe algorithm [12] that globally optimizes for the permutations of all the layers simultaneously;
- We generalize it to the case of $n \geq 2$ models, enforcing guaranteed cycle-consistency of the permutations by employing a universal space as a bridge;
- We leverage the multi-model matching procedure for model merging, using the universal space as aggregation point;
- We conduct an extensive analysis showing how the merge is affected by the number of models, their width and architecture, as well as quantitatively measuring the linear mode connectivity in the universe basin.

Finally, to foster reproducible research in the field, we release a modular and reusable codebase containing implementations of our approach and the considered baselines.[1]

---

[1] https://github.com/crisostomi/cycle-consistent-model-merging

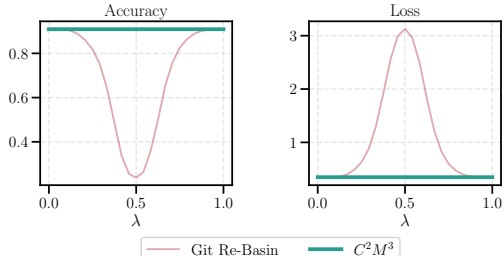

| Permutation | Git Re-Basin | $C^2M^3$ |
|---|---|---|
| $d\left(A, P_{A\rightarrow B\rightarrow C\rightarrow A}(A)\right)$ | 41.07 | 0.0 |
| $d\left(B, P_{B\rightarrow C\rightarrow A\rightarrow B}(B)\right)$ | 41.18 | 0.0 |
| $d\left(C, P_{C\rightarrow A\rightarrow B\rightarrow C}(C)\right)$ | 41.19 | 0.0 |

(a) Loss and accuracy curves for a model $A$ and the model mapped back after a cyclic permutation. Models cyclically permuted with `Git Re-Basin` end up in a different basin than the one they started from.

(b) Accumulated error obtained when cyclically permuting models $A$, $B$ and $C$ as in Figure 1. $P_{A\rightarrow B\rightarrow C\rightarrow A}$ refers to the composition $P_{AC} \circ P_{CB} \circ P_{BA}$ and $d(\cdot)$ is the $\ell_2$ loss.

Figure 2: Existing methods accumulate error when cyclically mapping a model through a series of permutations, while $C^2M^3$ correctly maps the model back to the starting point.

## 2    Background

**Mode connectivity**    As introduced in Section 1, mode connectivity studies the geometry of the loss landscape with a particular interest on the regions corresponding to local minima. Following Frankle et al. [13], we assess the connectivity for two given modes by computing their loss barrier:

**Definition 2.1.** (*Loss barrier*) Given two points $\Theta_A, \Theta_B$ and a loss function $\mathcal{L}$ such that $\mathcal{L}\left(\Theta_A\right) \approx \mathcal{L}\left(\Theta_B\right)$, the *loss barrier* is defined as

$$\max_{\lambda \in [0,1]} \mathcal{L}\left((1-\lambda)\Theta_A + \lambda\Theta_B\right) - \frac{1}{2}\left(\mathcal{L}\left(\Theta_A\right) + \mathcal{L}\left(\Theta_B\right)\right).$$

Intuitively, this quantity measures the extent of the loss increase when linearly moving from the basin of a mode to the other. When two modes share the same basin, the loss does not increase at all and results in a barrier close to zero.

**Weight-space symmetries**    Following the rich line of works on mode connectivity and model merging [1, 11, 13, 29, 36], we start from the essential insight of *neuron permutation invariance* in neural networks. Let us focus on the simple case of a Multi-Layer Perceptrons (MLP), where we can write the computation for an intermediate layer $W_\ell \in \mathbb{R}^{d_{\ell+1} \times d_\ell}$ as $z_{\ell+1} = \sigma\left(W_\ell z_\ell + b_\ell\right)$, with $z_\ell$ being the input at the $\ell$-layer and $\sigma$ denoting an element-wise activation function. For the sake of a clear exposure, we consider the bias $b_\ell = 0$ in the following. If apply a permutation matrix $P \in \mathbb{P}$ to the rows of the $W_\ell$ matrix (*i.e.* the neurons), we obtain $z'_{\ell+1} = \sigma\left(PW_\ell z_\ell\right)$. Being an element-wise operator, $\sigma$ commutes with $P$ and can be neglected wlog. Since $z'_{\ell+1} \neq z_\ell$ when $P \neq I$, we can still nullify the effect of the permutation by anti-permuting the columns of the subsequent layer for the inverse permutation of $P$, *i.e.* $P^\top$. In fact,

$$z'_{\ell+2} = W_{\ell+1}P^\top z'_{\ell+1} = W_{\ell+1}\underbrace{P^\top P}_{I}W_\ell z_\ell = z_{\ell+2}$$

making pairs of models that only differ by a permutation of the neurons de facto functionally equivalent. Given the enormous number of such permutations, it stands to reason that the resulting weight-space symmetries act as a major factor in the isolation of modes.

**Solving for the permutation**    Given the above considerations, Entezari et al. [11] speculated that all models end up in a single basin after having accounted for permutation symmetries. Assuming this to hold at least in practical cases, Ainsworth et al. [1] proposed a simple algorithm to find the permutations matching two models by maximizing a local version of the sum of bi-linear problems:

$$\arg\max_{\{P_\ell \in \mathbb{P}\}} \sum_{\ell=1}^{L} \langle W_\ell^A, P_\ell W_\ell^B P_{\ell-1}^T \rangle, \tag{1}$$

with $P_0 := I$. Noting that Equation (1) is NP-hard, Ainsworth et al. [1] tackle this problem by considering one layer at a time, relaxing the bi-linear problems to a set of linear ones that can be efficiently solved with any Linear Assignment Problem (LAP) solver, e.g., the Hungarian algorithm. This layer-wise linearization of the objective function, however, corresponds to high variance in the results that depend on the random order of the layers during optimization. See Table 7 for an empirical evaluation confirming this issue.

**Renormalizing the activations**    Notwithstanding the quality of the obtained matching, the loss barrier can still be high due to the mismatch in the statistics of the activations. In fact, REPAIR [21] empirically shows the presence of a decay in the variance of the activations after the interpolation. They further show that the loss can be drastically reduced by "repairing" the mean and variance of the activations, forcing the statistics of the merged network to interpolate those of the endpoint networks. We refer the reader to Appendix A.4 for an in-depth explanation.

## 3    Approach

We now propose a novel algorithm to tackle the weight matching problem, first introducing its formulation in the pairwise case and then generalizing it to match and merge a larger number $n$ of models in a cycle-consistent fashion.

**Pairwise matching**    As we have seen, the NP-hardness of Equation (1) demands for a relaxation of the problem to be tackled. Differently from Ainsworth et al. [1], we opt to maintain the objective global with respect to the layers and instead iteratively optimize its linear approximation via the the Frank-Wolfe algorithm [12]. This procedure requires the computation of the gradient of Equation (1) with respect to each permutation $P_i$, thus we have to account for two contributions for each $\nabla_{P_i}$, *i.e.*, its gradient from permuting the rows of $W_i$ and the one from permuting the columns of $W_{i+1}$:

$$\nabla_{P_i} f = \underbrace{W_i^A P_{i-1} (W_i^B)^\top}_{\text{from permuting rows}} + \underbrace{(W_{i+1}^A)^\top P_{i+1} W_{i+1}^B}_{\text{from permuting columns}}. \tag{2}$$

The Frank-Wolfe algorithm then uses the gradient to iteratively update the solution by linearly interpolating between the current solution and the projected gradient. We refer to Lacoste-Julien [24] for theoretical guarantees of convergence. The full algorithm is reported in Appendix A.2.

**Generalization to $n$ models**    In order to generalize to $n$ models, we jointly consider all pairwise problems

$$\arg \max_{P_i^{pq} \in \mathbb{P}} \sum_{p=1}^{n} \sum_{\substack{q=1 \\ q \neq p}}^{n} \sum_{i=1}^{L} \langle W_i^p, P_i^{pq} W_i^q (P_{i-1}^{pq})^\top \rangle, \tag{3}$$

where the superscript $pq$ indicates that the permutations maps model $q$ to model $p$, with $P_0^{pq} := I$. In order to *ensure cycle consistency by construction* we replace the quadratic polynomial by a fourth-order polynomial. Dropping the layer subscript for the sake of clear exposure, we replace the pairwise matchings $P^{pq}$ in the objective of Equation (3) by factorizing the permutations into *object-to-universe matchings* $P^{pq} = P^p \circ (P^q)^\top$ so that each model $q$ can be mapped back and forth to a common universe $u$ with a permutation and its transpose, allowing to map model $q$ to model $p$ by composition of $(P^q)^\top$ ($q \to u$) and $P^p$ ($u \to p$). This way, the objective of Equation (3) becomes

$$\sum_{p \neq q}^{n} \sum_{i=1}^{L} \langle W_i^p, P_i^p (P_i^q)^\top W_i^q (P_{i-1}^p (P_{i-1}^q)^\top)^\top \rangle = \sum_{p \neq q}^{n} \sum_{i=1}^{L} \langle (P_i^p)^\top W_i^p P_{i-1}^p, (P_i^q)^\top W_i^q P_{i-1}^q \rangle. \tag{4}$$

As stated by Theorem 3.1, the permutations we obtain using Equation (4) are cycle consistent. We refer the reader to Bernard et al. [5] for the proof and a complete discussion of the subject.

**Theorem 3.1.** *Given a set of $n$ models $p_0, \dots, p_n$ and object-to-universe permutations $P_i^{p_j}$ computed via Equation (4), the pairwise correspondences defined by $P_i^{p_l p_j} = P_i^{p_l} \circ (P_i^{p_j})^T$ are cycle-consistent, i.e.,*

$$P_i^{p_1 p_j} \circ \cdots \circ P_i^{p_3 p_2} \circ P_i^{p_2 p_1} = I$$

*for all layer indices $i$, $2 \leq j \leq n$.*

Similarly to the pairwise case, the approach requires computing the gradients for the linearization. This time, however, each $\nabla_{P_i^A} f$ has four different contributions: one from permuting the rows of its corresponding layer, one from anti-permuting the columns of the subsequent layer, and two other contributions that arise from the symmetric case where $A$ becomes $B$. In detail,

$$\nabla_{P_\ell^A} = \nabla_{P_\ell^A}^{\text{rows}} + \nabla_{P_\ell^A}^{\text{cols}} + \nabla_{P_\ell^A}^{\text{rows},\leftrightarrows} + \nabla_{P_\ell^A}^{\text{cols},\leftrightarrows} \qquad (5)$$

where

$$\nabla_{P_\ell^A}^{\text{rows}} = W_\ell^A P_{\ell-1}^A (P_{\ell-1}^B)^\top (W_\ell^B)^\top P_\ell^B \qquad \nabla_{P_\ell^A}^{\text{cols}} = (W_{\ell+1}^A)^\top P_{\ell+1}^A (P_{\ell+1}^B)^\top W_{\ell+1}^B P_\ell^B$$

$$\nabla_{P_\ell^A}^{\text{rows},\leftrightarrows} = W_\ell^B P_{\ell-1}^B (P_{\ell-1}^A)^\top (W_\ell^A)^\top P_\ell^A \qquad \nabla_{P_\ell^A}^{\text{cols},\leftrightarrows} = (W_{\ell+1}^B)^\top P_{\ell+1}^B (P_{\ell+1}^A)^\top W_{\ell+1}^A P_\ell^A$$

See Algorithm 1 for a complete description of the procedure.

---

**Algorithm 1** Frank-Wolfe for $n$-Model Matching

---

**Require:** Weights of $n$ models $M_{i=1}^N$
     tolerance $\epsilon > 0$
**Ensure:** Approximate solution to Equation (4)
1:  $\mathbf{P}^k \leftarrow$ identity matrices
2:  **repeat**
3:     **for** $(p,q) \in [1,\ldots,n] \times [1,\ldots,n]$ **do**
4:        **for** $i = 1$ to $L$ **do**
5:           $P_i^{p,k}, P_{i-1}^{p,k} \leftarrow$ permutations over rows and columns of $W_i^p$ respectively
6:           $P_i^{q,k}, P_{i-1}^{q,k} \leftarrow$ permutation over rows and columns of $W_i^q$ respectively
7:           $\nabla_{P_i^{p,k}} f \pm (W_{\ell+1}^p)^\top P_{\ell+1}^p (P_{\ell+1}^q)^\top W_{\ell+1}^q P_\ell^q$
8:           $\nabla_{P_{i-1}^{p,k}} f \pm (W_{\ell+1}^p)^\top P_{\ell+1}^p (P_{\ell+1}^q)^\top W_{\ell+1}^q P_\ell^q$
9:        **end for**
10:     **end for**
11:     **for** $P_i^k \in \mathbf{P}^k$ **do**
12:        $\Pi_i \leftarrow \text{LAP}(\nabla_{P_i^K} f)$
13:     **end for**
14:     $\alpha \leftarrow$ line search$(f, \mathbf{P}^k, \mathbf{\Pi})$
15:     **for** $P_i^k \in \mathbf{P}^k$ **do**
16:        $P_i^{k+1} = (1-\alpha) P_i^k + \alpha \, \Pi_i$
17:     **end for**
18: **until** $\|f(A, B, \mathbf{P}^{k+1}) - f(A, B, \mathbf{P}^k)\| < \epsilon$
19: **return** $\mathbf{P}^k$

---

**Merging in the universe space**   Looking at the loss landscape resulting from interpolating models in Figure 3, we see that the loss curves are much lower when the models are interpolated in the universe space. In fact, the originally disconnected modes end up in the same basin when mapped onto the universe, making it suitable to average the models. Therefore, our merging method aggregates the models by taking the mean of the weights in the universe space, as detailed in Algorithm 2.

---

**Algorithm 2** $C^2 M^3$: Cycle-Consistent Multi Model Merging

---

**Require:** $N$ models $A_1, \ldots, A_N$ with $L$ layers
**Ensure:** merged model $M$
1:  $\{P_1, \ldots, P_N\} \leftarrow$ Frank-Wolfe$(M_1, \ldots, M_N)$
2:  **for** $i = 1$ to $N$ **do**
3:     $M_i^{\text{uni}} \leftarrow$ map_to_universe$(A_i, P_i)$
4:  **end for**
5:  $M^{\text{uni}} \leftarrow \frac{1}{N} \sum_{i=1}^N M_i^{\text{uni}}$
6:  **return** $M^{\text{uni}}$

---

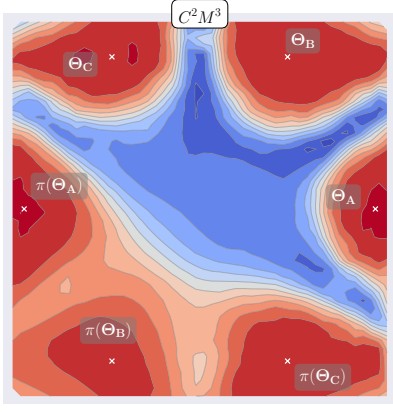

(a) `ResNet20` over `CIFAR100`.

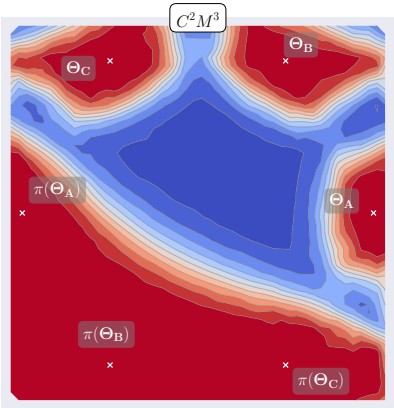

(b) `MLP` over `MNIST`.

Figure 3: 2D projection of the loss landscape when matching three modes $\Theta_A, \Theta_B, \Theta_C$; the models $\pi(\Theta_A), \pi(\Theta_B), \pi(\Theta_C)$ are their resulting images in the universe, and lie in the same basin. Red zones indicate low-loss regions (typically basins), while blue zones indicate high-loss ones.

## 4 Experiments

We now evaluate the quality of our proposed framework both in matching models and in the subsequent merging operation. Approaches suffixed with a † indicate the application of `REPAIR`.

**Matching and merging two models** As described in Section 3, our formalization can readily be used to match $n = 2$ models. In this case, the energy is given by Equation (1) and the permutations are not factorized. We compare the performance of our approach against the `Git Re-Basin` algorithm [1] and the `naive` baseline that aggregates the models by taking an unweighted mean on the original model weights without applying any permutation. In this setting, our method performs on par with the state-of-the-art. Differently from the latter, however, we do not depend on the random choice of layers, as the optimization is performed over all layers simultaneously. As presented in Figure 4, this results in `Git Re-Basin` exhibiting variations of up to $10\%$ in accuracy depending on the optimization seed, while our method shows zero variance. We refer the reader to Appendix B.1 for a thorough evaluation of $C^2M^3$ over a set of different datasets

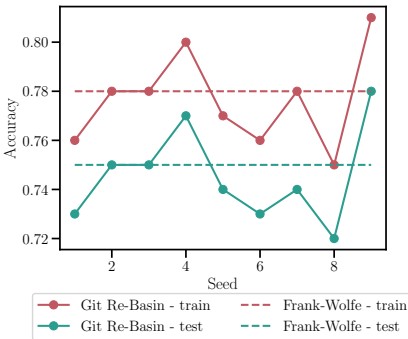

Figure 4: Accuracy of the interpolated model using `Git Re-Basin` [1] over different optimization seeds.

and architectures. In summary, *our approach is able to match two models with the same accuracy as the state-of-the-art, while being deterministic and independent of the random choice of layers.*

**Matching and merging $n$ models** We now evaluate $C^2M^3$ in matching and merging $n$ models. The matching is given by the factorized permutations obtained by Algorithm 1. We compare against two baselines: the simple approach of naively averaging the weights without any matching, and the `MergeMany` approach proposed by Ainsworth et al. [1]. The latter is reported in Appendix A for convenience. As reported in Table 1, $C^2M^3$ obtains far superior results in terms of accuracy and loss in all considered settings, with accuracy gains as high as $+20\%$. Moreover, our approach natively yields cycle-consistent permutations: Figure 2b shows that `Git Re-Basin` [1] accumulates significant error when computing the distance between the source model and the model obtained by applying a cyclic series of permutations, while our approach is able to perfectly recover the source model. This is further confirmed in Figure 2a, where we show the loss and accuracy curves when interpolating between a model $A$ and the model mapped back after a cyclic permutation.

| Matcher | | EMNIST | | | | | CIFAR10 | | | | | CIFAR100 | | | |
|---|---|---|---|---|---|---|---|---|---|---|---|---|---|---|---|
| | | **Accuracy** (↑) | | **Loss** (↓) | | | **Accuracy** (↑) | | **Loss** (↓) | | | **Accuracy** (↑) | | **Loss** (↓) | |
| | | train | test | train | test | | train | test | train | test | | train | test | train | test |
| Naive | MLP | 0.03 | 0.03 | 3.28 | 3.28 | ResNet 2× | 0.10 | 0.10 | 3.07 | 3.07 | ResNet 4× | 0.01 | 0.01 | 5.30 | 5.30 |
| MergeMany | | 0.88 | 0.86 | 1.11 | 1.13 | | 0.38 | 0.37 | 2.08 | 2.06 | | 0.31 | 0.28 | 3.01 | 2.76 |
| MergeMany† | | 0.88 | 0.86 | 1.11 | 1.13 | | 0.50 | 0.50 | 2.34 | 2.30 | | 0.24 | 0.22 | 3.31 | 3.12 |
| $C^2M^3$ | | 0.89 | 0.87 | 1.07 | 1.10 | | 0.42 | 0.40 | 2.11 | 2.05 | | 0.34 | 0.30 | 2.94 | 2.63 |
| $C^2M^{3\dagger}$ | | **0.89** | **0.87** | **1.07** | **1.10** | | **0.72** | **0.69** | **1.26** | **1.12** | | **0.53** | **0.46** | **2.13** | **1.67** |
| Naive | ResNet 2× | 0.04 | 0.04 | 4.04 | 4.04 | VGG16 | 0.10 | 0.10 | 2.31 | 2.31 | ResNet 16× | 0.01 | 0.01 | 6.22 | 6.22 |
| MergeMany | | 0.03 | 0.03 | 7.17 | 7.18 | | 0.10 | 0.10 | 2.36 | 2.36 | | 0.45 | 0.38 | 2.32 | 3.06 |
| MergeMany† | | 0.03 | 0.03 | 4.74 | 4.72 | | 0.60 | 0.57 | 1.43 | 1.32 | | 0.41 | 0.35 | 2.27 | 2.68 |
| $C^2M^3$ | | 0.27 | 0.27 | 3.43 | 3.47 | | 0.11 | 0.11 | 2.34 | 2.34 | | 0.46 | 0.39 | 2.25 | 3.03 |
| $C^2M^{3\dagger}$ | | **0.60** | **0.60** | **1.32** | **1.34** | | **0.64** | **0.62** | **1.34** | **1.23** | | **0.60** | **0.49** | **1.43** | **2.23** |

Table 1: Accuracy of the merged model when merging 5 models trained with different initializations. The best results are highlighted in bold. † denotes models after the REPAIR operation.

Models cyclically permuted with Git Re-Basin end up in a different basin than the one they started from, while our cycle-consistent approach ensures that the target model is exactly the same as the source. Wrapping up, *our approach matches and merges $n$ models with a significant improvement in performance over the state-of-the-art, while ensuring cycle-consistent permutations.*

**Model similarity before and after mapping**    As we can see in Figure 5, the cosine similarity of the weights of the models is 3× higher after mapping the latter to the universe. This suggests that the initial quasi-orthogonality of models is at least partially due to neuron permutation symmetries. We also report in Appendix C.1.2 the similarity of the representations between pairs of models. Interestingly, the latter does not change before and after mapping to the universe, but only if we consider a similarity measure that is invariant to orthogonal transformations such as CKA [22]. When using a measure that does not enjoy this property, such as the Euclidean distance, the representations become much more similar in the universe space. In short, *the models are 3× more similar in the universe space and the mapping affects the representations as an orthogonal transformation.*

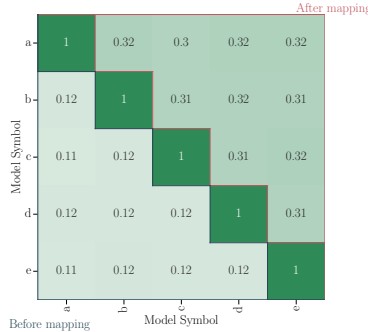

Figure 5: Cosine similarity of the weights of 5 `ResNet20` trained on `CIFAR10` with 2× width.

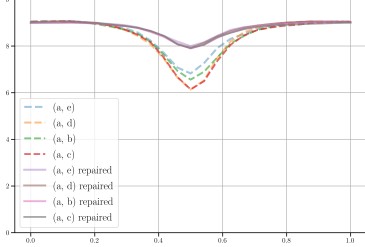

Figure 6: Interpolation curves of VGG models in the universe.

**Effect of activation renormalization**    Our empirical evidence also points out the benefits of the REPAIR operation [21] that is performed after the merging. In fact, the detrimental effect of model averaging on the activation statistics [21] still applies when taking the mean of $n$ models instead of two. Our results clearly show the benefit of REPAIR, making it a key ingredient of our overall framework. Requiring meaningful interpolation endpoints to be effective, REPAIR has lower benefit when employed on the MergeMany algorithm of Ainsworth et al. [1]. In fact, iteratively taking means of different random model subsets and aligning the left-out models to the mean is a more complex process than interpolating between some endpoint models. By taking the mean of models in the universe space, we are instead effectively interpolating between endpoint models that can be used for the computation of the statistics in Equation (8). Figure 6 shows the benefit of using the repair operation on 5 VGG models trained on CIFAR10 mapped to the universe space. Specifically we fix one model "a" and we linearly interpolate in the universe space with respect to the other models, measuring accuracy before and after applying REPAIR. Other than boosting performance, we observe that the latter reduces the variance over interpolation paths, resulting in the interpolation curves of all the models overlapping. Overall, *using the models in the universe as meaningful endpoints to gather activation statistics, our approach can fully leverage activation renormalization techniques such as REPAIR.*

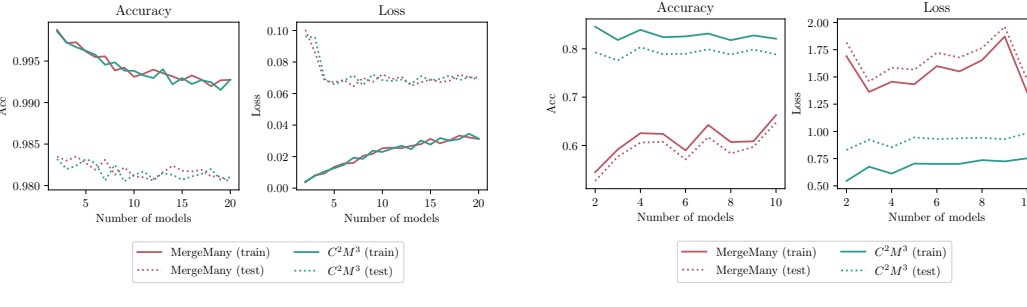

(a) MLPs trained over `MNIST`.

(b) `ResNet20` models trained over `CIFAR10`.

Figure 7: Accuracy and loss when increasing the number $n$ of models to match and merge.

**Increasing $n$** In this experiment, we show how the merged model behaves when increasing the number of aggregated models. As we can see in Figure 7a, increasing the number of MLPS up to 20 causes the performance to slightly deteriorate in a relative sense, but remaining stable in an absolute sense as it doesn't fall below $98\%$. More surprisingly, Figure 7b shows that for a `ResNet20` architecture with $4\times$ width the loss and accuracy are not monotonic, but rather they seem to slightly fluctuate. This may hint at the merging process being more influenced by the composition of the model set, than by its cardinality. Intuitively, a model that is difficult to match with the others will induce a harder optimization problem, possibly resulting in a worse merged model. We dive deeper in the effect of the composition of the set of models in Appendix C.2. In short, *our approach is effective in merging a larger number of models, suggesting promise in federated settings.*

**Varying widths** We now measure how architectural width affects model merging, taking into consideration `ResNet20` architectures with width $W \in \{1, 2, 4, 8, 16\}$. As we can see in Figure 8, *width greatly increases the performance of the merged model*, reaching the zero-loss barrier first observed in [1] when $W = 16$. This is in line with the observations relating linear mode connectivity and network widths [11, 1], and confirms the intuition that the merging is only effective when modes *can* be linearly connected.

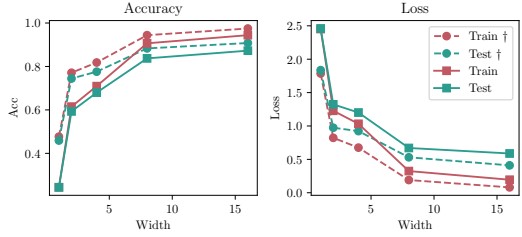

Figure 8: Accuracy and loss when merging 3 `ResNet20`s trained over `CIFAR10` with different widths. † indicates models after applying `REPAIR`.

**Alternative: fixing one model as universe** Alternatively, one could achieve cycle consistency by using one of the source models as reference and learning pairwise maps towards this one. This, however, would require arbitrarily selecting one of the models, making the overall merging dependent on an arbitrary choice. To see why this matters, we merged 5 `ResNet20-4×` by choosing one model as reference and aggregating the models in its basin. Figure 9 shows severe oscillations in the results, with one model reaching an accuracy as low as 65%, while our approach performs as the best possible reference. This approach, moreover, does not address multi-model merging, as it is intrinsically pairwise: in a multi-task setting, models optimally mapped to a reference basin would only be able to solve the task solved by the reference model. This would prevent merging to be used for models containing complementary information, such as knowledge fusion [19]

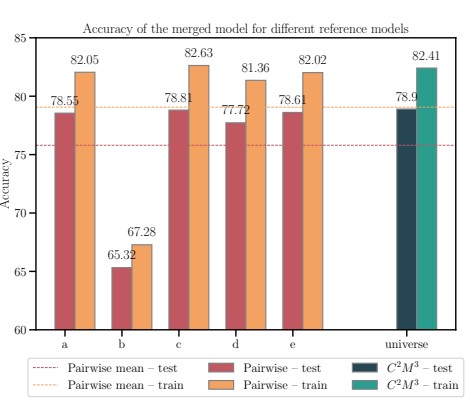

Figure 9: Accuracy of the merged model when mapping towards one arbitrary model (a, b, c, d, e) versus using $C^2 M^3$ and the universe space.

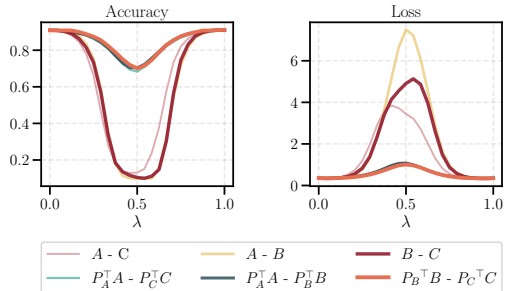
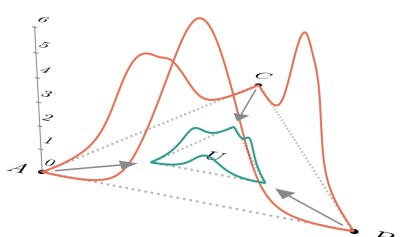

(a) 2D visualization of accuracy and loss of the models sampled from the pairwise interpolation lines.

(b) 3D visualization of the loss of the models sampled from the pairwise interpolation lines.

Figure 10: Linear mode connectivity before and after mapping to the universe for 3 `ResNet20-2×` models trained over `CIFAR10` according to Algorithm 1.

or multi-task merging [36]. In our setting, instead, the universe model must by design be a function of all the models and act as a midpoint, hence aggregating information from all the models.

**Linear mode connectivity in the universe**   Figure 10 shows that the loss curves of models interpolated in the universe are much lower than those interpolated in the original space, suggesting that the models are more connected in the former. These results, together with the loss landscape observed in Figure 3, *encourage merging the models in the universe space due to the lower loss barrier.*

## 5   Related work

**Mode Connectivity and model merging.** Mode connectivity studies the weights defining local minima. Frankle et al. [13] studied linear mode connectivity of models that were trained for a few epochs from the same initialization and related it to the lottery ticket hypothesis. Without requiring the same initialization, Entezari et al. [11] speculated that all models share a single basin after having solved for the neuron permutations. Model merging aims at aggregating different models into a single one to inherit their capacities without incurring in the cost and burden of ensembling. In this regard, Singh and Jaggi [34] proposed an optimal-transport based weight-matching procedure, while `Git Re-Basin` [1] proposed three matching methods and the `MergeMany` procedure seen in Section 4. Subsequently, `REPAIR` [21] showed that a significant improvement in performance of the interpolated model may be obtained by renormalizing its activations rather than changing matching algorithm. Differently from all these works, we consider merging $n$ models and propose a principled way to perform it with cycle-consistency guarantees.

**Cycle consistency.** Ensuring cycle consistency of pairwise maps is a recurring idea in the computer vision and pattern recognition literature. In the realm of deep learning, earlier studies addressing multi-graph matching achieved cycle consistency by synchronizing ex-post the predicted pairwise permutations [40, 41]. The alternative approach using an object-to-universe matching framework, which we adopt here, inherently ensures cycle consistency by construction, as demonstrated in [4, 16, 31]. To the best of our knowledge, none of the existing works tackles cycle-consistent alignment of neural models. We refer to Appendix A.1 for a more detailed list of related works.

## 6   Conclusions

In this work, we treated the problem of model matching and merging. We first introduced a novel weight matching procedure based on the Frank-Wolfe algorithm that optimizes for the permutation matrices of all layers jointly, and then generalized it to the case of $n$ models. Guaranteeing cycle-consistency, the latter poses a principled way to merge a set of models without requiring an arbitrary reference point. We then showed the approach to yield superior performance compared to existing ones in merging multiple models in a set of scenarios spanning different architectures and datasets. We believe the formalism to elegantly fit the requirement for the merging operation to unify the different models into a cohesive one, rather than mapping all of them to one of the models in the set.

## Acknowledgments

This work is supported by the ERC grant no.802554 (SPECGEO), PRIN 2020 project no.2020TA3K9N (LEGO.AI), and PNRR MUR project PE0000013-FAIR. Marco Fumero is supported by the MSCA IST-Bridge fellowship which has received funding from the European Union's Horizon 2020 research and innovation program under the Marie Skłodowska-Curie grant agreement No 101034413. We thank Simone Scardapane for the helpful feedback on the paper.

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

# Contents

# A    Additional details

Here we report in-depth explanations and additional experimental details. In particular, Appendix A.1 extensively outlines the most related works, Appendix A.2 shows the `Frank-Wolfe` algorithm for the pairwise case, while Appendix A.3 describes the `MergeMany` procedure presented in [1] for merging multiple models. We also report the `REPAIR` method in Appendix A.4. Finally, we show how the matching algorithm empirically converges in Appendix A.5.

## A.1    Extended related work

We report here a thorough review of works that are relevant to our research, providing a comprehensive understanding of the context of our work.

**Linear mode connectivity**    Mode connectivity is interested in modes, *i.e.* model parameters at convergence. In this regard, Frankle et al. [13] first studied the connectivity of the parameters of models that were trained for a few epochs from the same initialization, while Garipov et al. [14] investigated whether these can be connected through a high-accuracy path without requiring the same initialization. Simultaneously, Draxler et al. [10] proposed an algorithm to find a *Minimum Energy Path* (MEP) between two modes of a neural network, showing that these paths are mostly flat in both the training and test landscapes. This implies that many minima actually live in a shared low-loss valley rather than in distinct basins. On a different perspective, Zhou et al. [43] proposed to study a class of neural functionals which are permutation-equivariant by design. Recent research proposes to study model behavior in the weight space beyond linear mode connectivity: Lubana et al. [26] show that different "mechanisms" in related models prevent simple paths of low loss in the weight space, while Zhou et al. [44] studied the linear connections between the linear features of each layer of differently trained models.

**Model merging**    Model merging [1, 29, 34, 20, 39, 36] has seen a surge of interest in the last years as a mean to ensemble models without incurring in the added computational cost. One of the first works in this direction is Singh and Jaggi [34], who proposed an optimal-transport based weight-matching procedure. Later, Ainsworth et al. [1] proposed three matching methods, one of which being data-free. Closer to our global optimization, Peña et al. [29] proposed a gradient-descent based procedure that iteratively updates soft permutation matrices maintaining their bistochasticity via a differentiable Sinkhorn routine. When the models to match have been trained on different tasks, Stoica et al. [36] introduce a more general "zip" operation that accounts for features that may be task-specific and further allow obtaining multi-headed models. Most recently, Navon et al. [27] proposed aligning models in the embedding space of a deep weight-space architecture. Finally, weight merging proved useful for large language models [20] and robotics [39]. For a complete survey of mode connectivity and model merging, we refer the reader to [25].

**Cycle consistency**    Cycle consistency is a recurrent idea in computer vision and pattern recognition, where it appears under different names (e.g., "synchronization", "loop constraints", or "multi-way matching") depending on the task. In the area of multi-view 3D reconstruction, Zach et al. [42] were probably the first to make an explicit attempt at finding solutions meeting the cycle-consistency requirement, although without ensuring theoretical guarantees on the result. In geometry processing, Cosmo et al. [8] ensured cycle-consistent alignment of collections of 3D shapes using an $n$-fold extension of the Gromov-Wasserstein distance with sparsity constraints. Overall, cycle consistency is a recurring idea in the computer vision [38, 42, 2] graph matching [28, 37, 32] and geometry processing literature [17, 8, 4].

## A.2    Pairwise Frank-Wolfe Algorithm

As introduced in Section 3, we optimize a layer-global objective by iteratively optimizing its linear approximation via the the Frank-Wolfe algorithm [12]. We compute the gradient of Equation (1) with respect to each permutation $P_i$, as the sum of two contributions for each $\nabla_{P_i}$: one from permuting the rows of $W_i$ and another from permuting the columns of $W_{i+1}$:

$$\nabla_{P_i} f = \underbrace{W_i^A P_{i-1}(W_i^B)^\top}_{\text{from permuting rows}} + \underbrace{(W_{i+1}^A)^\top P_{i+1} W_{i+1}^B}_{\text{from permuting columns}}. \tag{6}$$

We report in Algorithm 3 the Frank-Wolfe algorithm for the pairwise case.

---

**Algorithm 3** Frank-Wolfe for pairwise Weight Matching

---

**Require:** Weights of two models $A$ and $B$ with $L$ layers, tolerance $\epsilon > 0$
**Ensure:** Approximate solution to Equation (1)
1: $\mathbf{P}^k \leftarrow$ identity matrices
2: **repeat**
3:     **for** $i = 1$ to $L$ **do**
4:         $P_i^k \leftarrow$ permutation acting on rows of $W_i$
5:         $P_{i-1}^k \leftarrow$ permutation acting on columns of $W_i$
6:         $\nabla_{P^K} f \mathrel{+}= W_i^A P_{i-1}^k (W_i^B)^\top$
7:         $\nabla_{P_{i-1}^k} f \mathrel{+}= (W_i^A)^\top P_i^k W_i^B$
8:     **end for**
9:     **for** $P_i^k \in \mathbf{P}^k$ **do**
10:       $\Pi_i \leftarrow \text{LAP}(\nabla_{P_i^K} f)$
11:     **end for**
12:     $\alpha \leftarrow \text{LINESEARCH}(f, \mathbf{P}^k, \mathbf{\Pi})$
13:     **for** $P_i^k \in \mathbf{P}^k$ **do**
14:       $P_i^{k+1} = (1 - \alpha) P_i^k + \alpha\,\Pi_i$
15:     **end for**
16: **until** $\|f(A, B, \mathbf{P}^{k+1}) - f(A, B, \mathbf{P}^k)\| < \epsilon$
17: **return** $\mathbf{P}^k$

---

## A.3 MergeMany Algorithm

Algorithm 4 reports the MergeMany procedure originally proposed by Ainsworth et al. [1] for merging multiple models, mainly consisting in alternating matching and aggregation until convergence. In practice, at each iteration, the procedure picks a reference model at random and matches all the other models to it. Then, they are all aggregated by averaging the weights.

---

**Algorithm 4** MERGEMANY

---

**Require:** Model weights $\Theta_1, \ldots, \Theta_N$
**Ensure:** A merged set of parameters $\tilde{\Theta}$.
1: **repeat**
2:     **for** $i \in \text{RANDOMPERMUTATION}(1, \ldots, N)$ **do**
3:         $\Theta' \leftarrow \frac{1}{N-1} \sum_{j \in \{1, \ldots, N\} \setminus \{i\}} \Theta_j$
4:         $\pi \leftarrow \text{PERMUTATIONCOORDINATEDESCENT}(\Theta', \Theta_i)$
5:         $\Theta_i \leftarrow \pi(\Theta_i)$
6:     **end for**
7: **until** convergence
8: **return** $\frac{1}{N} \sum_{j=1}^N \Theta_j$

---

## A.4 REPAIR

Observing a decay in the variance of the activations of the aggregated model, Jordan et al. [21] proposed REPAIR, which renormalizes the activations of the merged model to match the statistics of the original models. In particular, given two endpoint models with activations $X_1$ and $X_2$, the activations $X_\alpha$ of the interpolated model are renormalized to have statistics:

$$\mathbb{E}\left[X_\alpha\right] = (1 - \alpha) \cdot \mathbb{E}\left[X_1\right] + \alpha \cdot \mathbb{E}\left[X_2\right] \tag{7}$$

$$\text{std}\left(X_\alpha\right) = (1 - \alpha) \cdot \text{std}\left(X_1\right) + \alpha \cdot \text{std}\left(X_2\right). \tag{8}$$

## A.5 Convergence and efficiency

We report here the convergence of our matching algorithm. In particular, Figure 11 shows the objective values during the optimization, exhibiting the expected monotonic increase, while Figure 12 shows the step sizes result of the line search at each iteration. Interestingly, Figure 12a shows that the step sizes are generally decreasing, but descend in an alternating manner. This is likely due to the fact that the permutations are obtained as consecutive interpolations, where even steps result in a soft permutation matrix that is the average of the current and next permutation matrix, while odd steps generally result in a hard permutation matrix with entries in $[0, 1]$. Figure 13 finally shows the intermediate permutation values during the optimization: at each step, the entries of the permutation matrix are the linear interpolation of the current solution and the projected gradient with factor $\alpha$ given by the step size. The red values in the figure represent entries currently being updated, which are neither 1 (blue) or 0 (yellow).

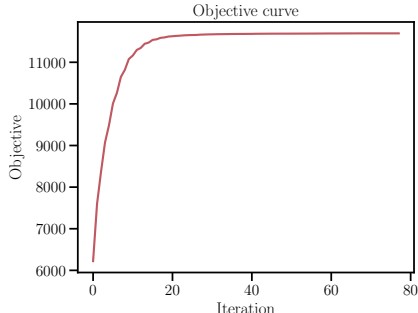

Figure 11: Objective values during the optimization. As guaranteed by the `Frank-Wolfe` algorithm, the objective value increases monotonically.

We report in Appendix A.5 the wall-clock time when merging $n = 2, 3$ `ResNet20` models having $1\times, 2\times, 4\times, 8\times$ and $16\times$ width, together with their number of parameters.

|  | 1x | 2x | 4x | 8x | 16x |
|---|---|---|---|---|---|
| # params | 292k | 1.166m | 4.655m | 18.600m | 74.360m |
| n=2 | | | | | |
| $C^2M^3$ | 33.4s | 33.5s | 40.5s | 80.8s | 367.8s |
| MergeMany | 0.24s | 0.4s | 3.4s | 8.9s | 59.4s |
| n=3 | | | | | |
| $C^2M^3$ time | 32.9s | 83.18s | 91.0s | 162.0s | 715.8s |
| MergeMany | 1.2s | 4.1s | 19.5s | 105.8s | 892.3s |

Table 2: Wall-clock time for merging $n = 3$ ResNet20 models with different widths.

As can be inferred from the table, the scaling laws depend on the complexity of the resulting matching problem and cannot be predicted merely from the number of parameters, with a 4-fold increase in parameters resulting in no increase in runtime for the first three columns, a double increase in the second-last column and a 5-fold increase in the last. Compared to MergeMany, our approach enjoys a milder increase in running time when increasing the number of parameters. For simpler settings, however, MergeMany is significantly faster. Being the two approaches on the same order of magnitude and given the one-time nature of model merging, we believe this aspect to be of secondary importance, especially considering merging to be, in many cases, an alternative to training a model from scratch.

## A.6 Architectural details

We report here the architectural details of all the architectures we have used in the experiments.

**Multi-Layer Perceptrons** We use a simple MLP mapping input to a 256-dimensional space followed by 3 hidden layers of 512, 512 and 256 units respectively, followed by an output layer mapping to the number of classes. We use *ReLU* activations for all layers except the output layer, where we use a *log_softmax* activation.

**ResNet** We consider a `ResNet20` [15] architecture composed by three ResNet block groups, each containing three residual blocks. The model starts with an initial convolutional layer followed by

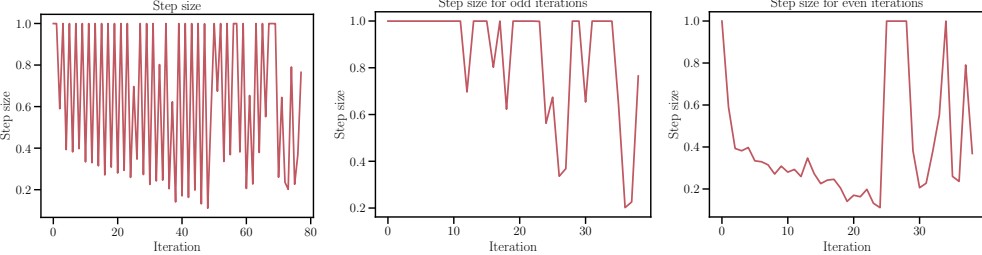

(a) Step sizes for all iterations.  (b) Step sizes for odd iterations.  (c) Step sizes for even iterations.

Figure 12: Step sizes during the optimization.

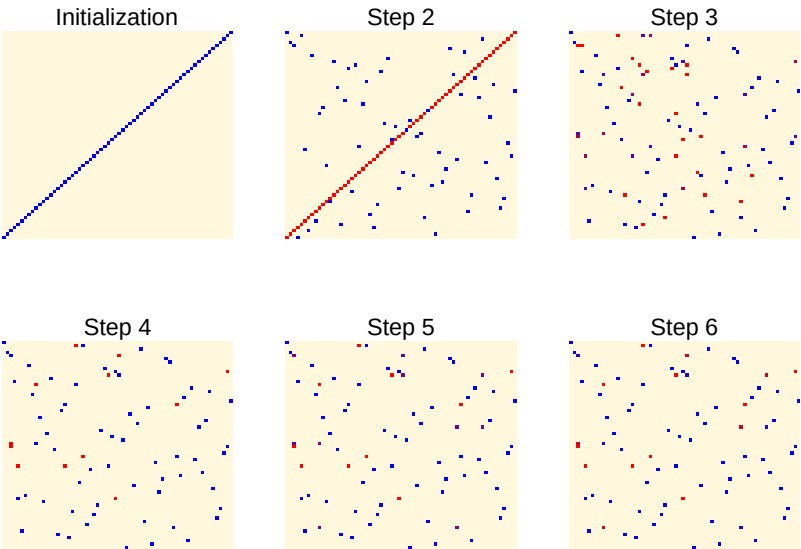

Figure 13: First 6 steps of Algorithm 3 for one permutation matrix. At each step, the new solution is given by the linear interpolation of the current solution and the gradient of Equation (1).

normalization and *ReLU* activation. It then passes through the three block groups with increasing channel sizes (determined by the widen factor) and varying strides, followed by global average pooling and a fully connected layer that outputs class logits. As normalization layers, we consider both the most commonly used *BatchNorm* [18] and, for the sake of comparing with `Git Re-Basin`, also *LayerNorm* [3]. The results in the main manuscript are all obtained with *LayerNorm*, while we report the results with *BatchNorm* in Appendix B.1.1.

**VGG**  We employ a `VGG16` [33] architecture with *LayerNorm* [3] normalization layers. The model has the following convolutional layer dimensions, with "M" indicating the presence of a max-pooling layer

$$64, 64, M, 128, 128, M, 256, 256, 256, M, 512, 512, 512, M, 512, 512, 512, M \qquad (9)$$

The convolutional layers are organized in 5 blocks, each containing 2 or 3 convolutional layers, followed by a max-pooling layer. The final classifier is composed of three fully connected layers with 512 hidden dimension and ReLU activations.

## A.7  Datasets, hyperparameters and hardware details

We employ the most common datasets for image classification tasks: `MNIST` [9], `CIFAR-10` [23], `EMNIST` [7] and `CIFAR-100` [23], having 10, 10, 26 and 100 classes respectively. We use the standard train-test splits provided by *torchvision* for all datasets.

We use the same hyperparameters as `Git Re-Basin` where possible to ensure a fair comparison. In particular, we train most of our models with a batch size of 100 for 250 epochs, using SGD with momentum 0.9, a learning rate of 0.1, and a weight decay of $10^{-4}$. We use a cosine annealing learning rate scheduler with a warm restart period of 10 epochs and a minimum learning rate of 0. We report each and every one of the hyperparameters used for each experiment, as well as all the trained models, in a WandB dashboard[2].

All of the experiments were carried out using consumer hardware, in particular mostly on a 32GiB RAM machine with a *12th Gen Intel(R) Core(TM) i7-12700F* processor and an *Nvidia RTX* 3090 GPU, except for some of the experiments that were carried on a 2080. Our modular and reusable codebase is based on *PyTorch*, leveraging *PyTorch Lightning* to ensure reproducible results and modularity and *NN-Template*[3] to easily bootstrap the project and enforce best practices.

## A.8 Proofs

**Theorem A.1.** *The gradient of the objective function*

$$\sum_{p=1}^{n-1} \sum_{q=p+1}^{n} \sum_{\ell=1}^{L} \langle (P_\ell^p)^\top W_\ell^p P_{\ell-1}^p, (P_\ell^q)^\top W_\ell^q P_{\ell-1}^q \rangle$$

*is Lipschitz continuous, implying our algorithm obtains a stationary point at a rate of $\mathcal{O}(1/\sqrt{t})$ [24].*

*Proof.* We recall that, for each layer permutation $P^A = \{P_1^A, P_2^A, \ldots, P_L^A\}$ of model $A$, we can define the gradient of our objective function relatively to the model $B$ we are matching towards:

$$f(P_\ell^A) = \nabla_{P_\ell^A}^{\text{rows}} + \nabla_{P_\ell^A}^{\text{cols}} + \nabla_{P_\ell^A}^{\text{rows},\leftrightarrows} + \nabla_{P_\ell^A}^{\text{cols},\leftrightarrows} =$$

$$\left[ W_\ell^A P_{\ell-1}^A (P_{\ell-1}^B)^\top (W_\ell^B)^\top + (W_{\ell+1}^A)^\top P_{\ell+1}^A (P_{\ell+1}^B)^\top W_{\ell+1}^B \right] P_\ell^B +$$

$$\left[ W_\ell^B P_{\ell-1}^B (P_{\ell-1}^A)^\top (W_\ell^A)^\top + (W_{\ell+1}^B)^\top P_{\ell+1}^B (P_{\ell+1}^A)^\top W_{\ell+1}^A \right] P_\ell^A$$

To prove Lipschitz continuity, we need to show there exists a constant $C$ such that $\forall \ p = 1, \ldots, n, \ \ell = 1, \ldots, L \quad \|f(P_\ell^p) - f(Q_\ell^p)\| \leq C \|P_\ell^p - Q_\ell^p\|$. To simplify passages, we only consider a fixed $\ell$ and perform a generic analysis. We begin by observing that

$$f(P_\ell^p) - f(Q_\ell^p) =$$

$$\sum_{q \in [1,n] \setminus \{p\}} \left[ W_\ell^p P_{\ell-1}^p (P_{\ell-1}^q)^\top (W_\ell^q)^\top + (W_{\ell+1}^p)^\top P_{\ell+1}^p (P_{\ell+1}^q)^\top W_{\ell+1}^q \right] (P_\ell^q - Q_\ell^q) +$$

$$\left[ W_\ell^q P_{\ell-1}^q (P_{\ell-1}^p)^\top (W_\ell^p)^\top + (W_{\ell+1}^q)^\top P_{\ell+1}^q (P_{\ell+1}^p)^\top W_{\ell+1}^p \right] (P_\ell^p - Q_\ell^p)$$

The last form of the above equation can be rewritten as a sum of the two sums:

$$\sum_{q \in [1,n] \setminus \{p\}} \left[ W_\ell^p P_{\ell-1}^p (P_{\ell-1}^q)^\top (W_\ell^q)^\top + (W_{\ell+1}^p)^\top P_{\ell+1}^p (P_{\ell+1}^q)^\top W_{\ell+1}^q \right] (P_\ell^q - Q_\ell^q) +$$

$$\sum_{q \in [1,n] \setminus \{p\}} \left[ W_\ell^q P_{\ell-1}^q (P_{\ell-1}^p)^\top (W_\ell^p)^\top + (W_{\ell+1}^q)^\top P_{\ell+1}^q (P_{\ell+1}^p)^\top W_{\ell+1}^p \right] (P_\ell^p - Q_\ell^p)$$

Since the first term does not depend on either $P_\ell^p$ or $Q_\ell^p$, we assume as a worst case that its norm is 0. Then, we remove transposes (since $\|M\| = \|M^\top\|$) and apply the triangle inequality and the sub-multiplicative property of matrix norms:

$$\|f(P_\ell^p) - f(Q_\ell^p)\| \leq$$

$$\sum_{q \in [1,n] \setminus \{p\}} \|P_\ell^p - Q_\ell^p\| \left( \|W_\ell^q\| \|P_{\ell-1}^q\| \|P_{\ell-1}^p\| \|W_\ell^p\| + \|W_{\ell+1}^q\| \|P_{\ell+1}^q\| \|P_{\ell+1}^p\| \|W_{\ell+1}^p\| \right)$$

Let $C = \max_{q \in [1,n] \setminus \{p\}} \left\{ \|W_\ell^q\| \|P_{\ell-1}^q\| \|P_{\ell-1}^p\| \|W_\ell^p\| + \|W_{\ell+1}^q\| \|P_{\ell+1}^q\| \|P_{\ell+1}^p\| \|W_{\ell+1}^p\| \right\}$. Then,

$$\|f(P_\ell^p) - f(Q_\ell^p)\| \leq C \sum_{q \in [1,n] \setminus \{p\}} \|P_\ell^p - Q_\ell^p\| = C(n-1) \|P_\ell^p - Q_\ell^p\|$$

---

[2]Link concealed to preserve anonymity.
[3]https://github.com/grok-ai/nn-template

| Matcher | Barrier | | | | | | | | | |
|---|---|---|---|---|---|---|---|---|---|---|
| | ResNet 2× | | ResNet 4× | | ResNet 8× | | ResNet 16× | | VGG16 | |
| | Train | Test | Train | Test | Train | Test | Train | Test | Train | Test |
| Naive | $5.16 \pm 1.83$ | $5.45 \pm 1.83$ | $2.94 \pm 0.27$ | $3.26 \pm 0.27$ | $2.12 \pm 0.03$ | $2.40 \pm 0.03$ | $1.84 \pm 0.18$ | $2.12 \pm 0.17$ | $1.85 \pm 0.00$ | $2.31 \pm 0.00$ |
| Git Re-Basin | $0.73 \pm 0.16$ | $0.86 \pm 0.17$ | $\mathbf{0.74 \pm 0.35}$ | $0.80 \pm 0.40$ | $0.19 \pm 0.03$ | $0.13 \pm 0.02$ | $0.17 \pm 0.02$ | $0.07 \pm 0.02$ | $0.08 \pm 0.03$ | $0.24 \pm 0.03$ |
| Frank-Wolfe | $0.73 \pm 0.19$ | $0.85 \pm 0.19$ | $0.78 \pm 0.33$ | $0.81 \pm 0.38$ | $0.19 \pm 0.03$ | $0.12 \pm 0.02$ | $0.16 \pm 0.02$ | $0.06 \pm 0.02$ | $0.08 \pm 0.03$ | $0.25 \pm 0.03$ |

Table 4: Mean and standard deviation of the test and train loss barrier for each method when matching $n = 2$ models on `CIFAR10`.

we conclude that $f(P_\ell^p)$ is Lipschitz continuous for all models and all layers, with Lipschitz constant $C(n - 1)$ depending on both the norm of the weights matrices and the number of models.

□

# B   Additional experiments

We report additional experiments and results in this section. In particular, Appendix B.1 presents a complete evaluation of our matching method for the pairwise case, showing it to be generally competitive with the state-of-the-art `Git Re-Basin` algorithm [1] and to outperform it on architectures employing *BatchNorm* [18] normalization. We then discuss different permutation initialization strategies in Appendix B.2.

## B.1   Pair-wise model matching and merging

| | Matcher | Barrier | |
|---|---|---|---|
| | | Train | Test |
| ResNet 8× | Naive | $7.00 \pm 1.24$ | $8.37 \pm 1.23$ |
| ResNet 8× | Git-Rebasin | $1.04 \pm 0.10$ | $1.54 \pm 0.13$ |
| ResNet 8× | Frank-Wolfe | $\mathbf{0.92 \pm 0.06}$ | $\mathbf{1.42 \pm 0.10}$ |
| VGG16 | Naive | $5.79 \pm 0.39$ | $7.36 \pm 0.38$ |
| VGG16 | Git-Rebasin | $0.44 \pm 0.03$ | $0.64 \pm 0.03$ |
| VGG16 | Frank-Wolfe | $\mathbf{0.44 \pm 0.05}$ | $\mathbf{0.63 \pm 0.06}$ |

Table 3: Mean and standard deviation of the test and train loss barriers for each method when matching $n = 2$ models on `CIFAR100`.

As described in Section 3, our formalization can readily be used to match $n = 2$ models. In this case, the energy is given by Equation (1) and the permutations are not factorized. We compare the performance of our approach against the `Git Re-Basin` algorithm [1] and the `naive` baseline that aggregates the models by taking an unweighted mean on the original model weights without applying any permutation. From the data presented in Table 3, we observe that the approach is competitive with `Git Re-Basin` [1], with the two methods exhibiting analogously low test barrier on `CIFAR10`. Focusing on the `ResNet20` architecture, we can see that width plays the same role in both approaches, with the barrier decreasing as it increases. We can also appreciate how, while the same architecture resulted in similar barriers for the two approaches on `CIFAR10`, the barrier is significantly lower for `Frank-Wolfe` in `CIFAR100`, possibly suggesting that the latter is more robust to the complexity of the dataset.

### B.1.1   ResNet with BatchNorm

We also report the results of a ResNet20 with $2\times$ width using *BatchNorm* [18] layers instead of *LayerNorm* [3] ones. This version, as noted in [21], is in fact harder to match but also the one that is commonly used in practice. We can see in Table 5 that the `Frank-Wolfe` matcher is able to achieve a lower barrier than `Git Re-Basin`, indicating the approach to be more robust to architectures using different normalization layers.

| Matcher | loss barrier ($\downarrow$) | |
|---|---|---|
| | train | test |
| Naive | $4.72 \pm 0.86$ | $4.99 \pm 0.86$ |
| Git Re-Basin | $4.33 \pm 0.64$ | $4.62 \pm 0.65$ |
| Frank-Wolfe | $\mathbf{3.53 \pm 0.58}$ | $\mathbf{3.79 \pm 0.57}$ |

Table 5: Mean and stddev of the test and train loss barriers on $2$ `ResNet20-2×` models with *BatchNorm* normalization.

## B.2   Initialization strategies

As introduced in Algorithm 1, we initialize each $N$-dimensional permutation to be the $N \times N$ identity matrix. We now compare this strategy against two alternatives that provide doubly stochastic

| models | | 1 | 2 | 3 | 4 | 5 | 6 | 7 | 8 | 9 | mean | stddev | max gap | Frank-Wolfe |
|---|---|---|---|---|---|---|---|---|---|---|---|---|---|---|
| (1,2) | train | 0.76 | 0.78 | 0.78 | 0.80 | 0.77 | 0.76 | 0.78 | 0.75 | 0.81 | 0.78 | 0.018 | 0.057 | 0.78 |
| | test | 0.73 | 0.75 | 0.75 | 0.77 | 0.74 | 0.73 | 0.74 | 0.72 | 0.78 | 0.74 | 0.018 | 0.060 | **0.75** |
| (1,3) | train | 0.67 | 0.69 | 0.69 | 0.69 | 0.62 | 0.69 | 0.66 | 0.71 | 0.68 | 0.68 | 0.023 | 0.085 | 0.68 |
| | test | 0.64 | 0.66 | 0.67 | 0.65 | 0.60 | 0.66 | 0.63 | 0.67 | 0.65 | 0.65 | 0.020 | 0.071 | 0.65 |
| (2,3) | train | 0.75 | 0.74 | 0.75 | 0.72 | 0.76 | 0.74 | 0.70 | 0.73 | 0.78 | 0.74 | 0.020 | 0.074 | **0.76** |
| | test | 0.70 | 0.71 | 0.71 | 0.68 | 0.72 | 0.70 | 0.67 | 0.70 | 0.74 | 0.70 | 0.020 | 0.071 | **0.72** |

Table 7: Accuracy of the interpolated model using `Git Re-Basin` [1] over different pairs of models $(1, 2), (1, 3), (2, 3)$ by changing random seed $i = 1, \ldots, 9$ in the weight matching procedure.

| models | loss barrier ($\downarrow$) | | |
|---|---|---|---|
| | id | barycenter | Sinkhorn |
| (a, b) | 0.52 | **0.47** | $0.60 \pm 0.04$ |
| (b, c) | 0.65 | 0.70 | $\mathbf{0.64 \pm 0.06}$ |
| (a, c) | 0.97 | 0.95 | $\mathbf{0.92 \pm 0.07}$ |

Table 6: Test barrier of the interpolations of 3 `ResNet20-2×` models using different initializations.

matrices, *i.e.*, such that their rows and columns sum to one: i) the Sinkhorn initialization [35] that initializes the permutation matrix as the solution of the Sinkhorn-Knopp algorithm [35]; ii) the barycenter of doubly stochastic matrices, *i.e.* the matrix where each element is given by $1/N$. Table 6 shows the test barrier of the interpolations of three `ResNet20-2×` models $a, b$, and $c$ when using the different strategies over 10 different trials. We can see that the constant initializations (identity and barycenter) work well in general, with the additional benefit of having 0 variance in the results. On the other hand, if computational cost is not a concern, one can still choose to run a pool of trials with different Sinkhorn initializations and finally select the best one, trading this way efficiency with some extra accuracy points.

## B.3 Variance of the results in Git Re-Basin

As introduced in Section 4, `Git Re-Basin` [1] depends on a random choice of layers, resulting in variations of up to $10\%$ in accuracy depending on the optimization seed, while our method shows zero variance. While we have already seen the results for a model pair in Figure 4, we report, for completeness, the results of matching and averaging models with `Git Re-Basin` using different optimization seeds for additional pairs. As can be seen in Table 7, the trend is confirmed over these ones, with results significantly oscillating and our approach always above or on par with their mean.

## B.4 Large-scale matching: ResNet50s trained over ImageNet

For this experiment, we matched three different `ResNet50s` trained over `ImageNet`. We used three publicly available pretrained checkpoints from *timm*, namely `a1_in1k`[4], `c1_in1k`[5] and `ram_in1k` [6]. As Table 8 shows, $C^2M^3$ underperforms the baseline in this case. To see why, we report in Figure 14 the pairwise accuracies obtained using pairwise weight matching over all the `ResNet50` checkpoints available in *timm*. Let us focus on the triplet (am, a2, ram) and replace the model names with (a, b, c) for clarity. We see that, while the mergings (a, b) and (b, c) result in high-accuracy models, the merging (a, c) yields poor results. Given the cycle consistency of our method, we inherit the difficulty of the hardest pair, which in this case is (a, c). It is worth noting that this behavior is not present in the other cases we investigated in this work, and might be due to

| Matcher | Accuracy ($\uparrow$) | Loss ($\downarrow$) |
|---|---|---|
| Naive | 0.001 | 6.91 |
| MergeMany | 0.001 | 6.91 |
| MergeMany$^\dagger$ | 0.30 | 4.87 |
| $C^2M^3$ | 0.001 | 6.91 |
| $C^2M^{3\dagger}$ | 0.07 | 6.13 |

Table 8: Accuracy and loss of the interpolated model using different matchers over three `ResNet50` models trained on `ImageNet`.

the considered models being trained with different training schedules and hyperparameters. Future research could investigate new strategies to handle such cases, *e.g.* by iteratively merging models by following a max-accuracy path in an accuracy weighted graph.

---

[4] https://huggingface.co/timm/resnet50.a1_in1k
[5] https://huggingface.co/timm/resnet50.c1_in1k
[6] https://huggingface.co/timm/resnet50.ram_in1k

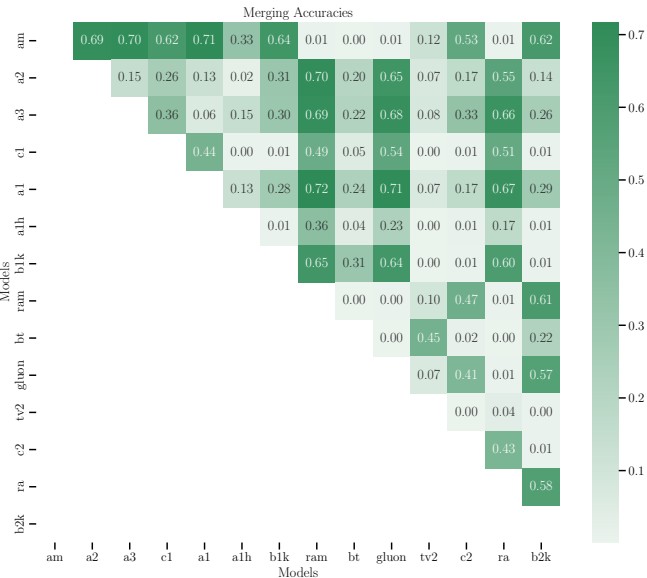

Figure 14: Pairwise accuracies obtained using `Git Re-Basin` [1] over different `ResNet50` models trained on `ImageNet`. Models are available from the *timm* library.

| Round | Accuracy | | | | | | | | | |
|---|---|---|---|---|---|---|---|---|---|---|
| | 1 | 5 | 10 | 15 | 20 | 25 | 30 | 35 | 40 | 45 |
| FedAvg | 0.0942 | 0.394 | 0.4972 | 0.5517 | 0.5699 | 0.5893 | 0.6018 | 0.6063 | 0.6099 | 0.6136 |
| $C^2M^3$ | 0.0941 | **0.4234** | **0.5193** | **0.5555** | **0.5783** | **0.5978** | **0.6077** | **0.6165** | **0.618** | **0.622** |

Table 9: Accuracy over 10 clients in a federated learning scenario. We report the accuracy for 50 aggregation rounds, with each client training for 20 local epochs. We report one every five rounds for the sake of clarity.

## B.5    Federated Learning

We here report the results of a preliminary experiment where we ran our framework in a federated learning scenario. To this end, we have used the state-of-the-art federated learning library Flower [7] [6] and employed our matching scheme over a set of 10 clients over `CIFAR10`, each adopting a small CNN model. We observe the following:

- When all the clients start from the same initialization, our approach has no benefit and falls back to standard averaging. In fact, the optimization process quickly returns identity matrices as permutations, suggesting the models already share the same basin.

- When instead we initialize the clients from different random initializations, Tables 9 and 10 show that our approach visibly outperforms FedAVG. In particular, the benefits get more pronounced when increasing the number of local epochs. This is in line with the intuition that standard averaging becomes less effective when clients drift due to prolonged local training and too infrequent aggregation.

While these results are not sufficient to claim an overall supremacy of the approach for the task due to the limited evaluation and choice of models, they show the approach to be promising for the problem and encourage further research.

---

[7] https://flower.ai/

| Round | Accuracy | | | | | | | | | |
|---|---|---|---|---|---|---|---|---|---|---|
| | 1 | 2 | 3 | 4 | 5 | 6 | 7 | 8 | 9 | 10 |
| FedAvg | 0.0942 | 0.2638 | 0.3543 | 0.3825 | 0.4165 | 0.4505 | 0.4742 | 0.4994 | 0.5169 | 0.5317 |
| $C^2M^3$ | **0.0947** | **0.3303** | **0.3899** | **0.4441** | **0.4764** | **0.4968** | **0.5184** | **0.5334** | **0.5434** | **0.5536** |

Table 10: Accuracy over 10 clients in a federated learning scenario. We report the accuracy for 10 aggregation rounds, with each client training for 30 local epochs.

## C Additional analysis

In this section, we report additional analyses that complement the results presented in the main text. We first analyze in Appendix C.1 how mapping to universe affects the similarity of the models; then, we evaluate how the composition of the match set affects the accuracy of the merged model in Appendix C.2.

### C.1 Similarity of models

We analyze here how similar are models before and after being mapped to the universe space, first by comparing their representations and then by comparing their weights.

#### C.1.1 Representation-level similarity

Figures 15a and 15b show the Centered Kernel Alignment (CKA) [22] of the representations of 5 `ResNet20` models trained on `CIFAR10` with $2\times$ width. The linear version of CKA is defined as

$$\text{CKA}(X,Y) = \frac{\text{HSIC}(X,Y)}{\sqrt{\text{HSIC}(X,X)\,\text{HSIC}(Y,Y)}}, \tag{10}$$

where $\text{HSIC}(X,Y) = \frac{1}{(N-1)^2} \text{tr}(\mathbf{XHX}^\top\mathbf{H})$, $\mathbf{H} = \mathbf{I} - \frac{1}{N}\mathbf{11}^\top$ is a centering matrix, and $\mathbf{1}$ is a vector of $N$ ones. The denominator is introduced to scale CKA between zero and one, where a value of one indicates equivalent representations. CKA is invariant to orthogonal transformations and isotropic scaling. Being permutations orthogonal transformations, CKA stays exactly the same after mapping the models to the universe. On the contrary, the Euclidean distance of the representations of the models significantly decreases after mapping to the universe, as shown in Figures 15c and 15d.

#### C.1.2 Weight-level similarity

We have seen in Figure 5 that the cosine similarity of the weights is higher after mapping the weights to the universe. This suggests that the models are more similar in the universe, which is consistent with the fact that it constitutes a convenient space to merge them. We report here for completeness the Figure 16 the Euclidean distance of the weights of 5 `ResNet20` models trained on `CIFAR10` with $2\times$ width, showing the same trend as the cosine similarity.

### C.2 Merging different subsets

We merge subsets of $k < 5$ models from the set of 5 models $a, b, c, d, e$ to gauge the effect of the match set composition over the accuracy of the merged model. As shown in Figure 17, we run two different merging schemes: in the former (left column), we globally match all the 5 models jointly and then consider subsets only at the aggregation step. In the second analysis (right column), we instead consider model subsets from the start, therefore running the whole matching procedure on the $k$ models before averaging them. This way, we aim to disentangle the error resulting from imperfect matching from the one naturally resulting from the aggregation. We highlight a few notable aspects:

1. While the accuracies are expectedly higher when matching a subset with permutations expressly optimized for that same subset (right column), this is not the case for $n = 2$, in which the permutations resulting from matching the superset of 5 models yield better results when merging pairs of them. This hints at the added constraint of cycle consistency over a wide number of models adding in some cases an advisable prior over the search space.

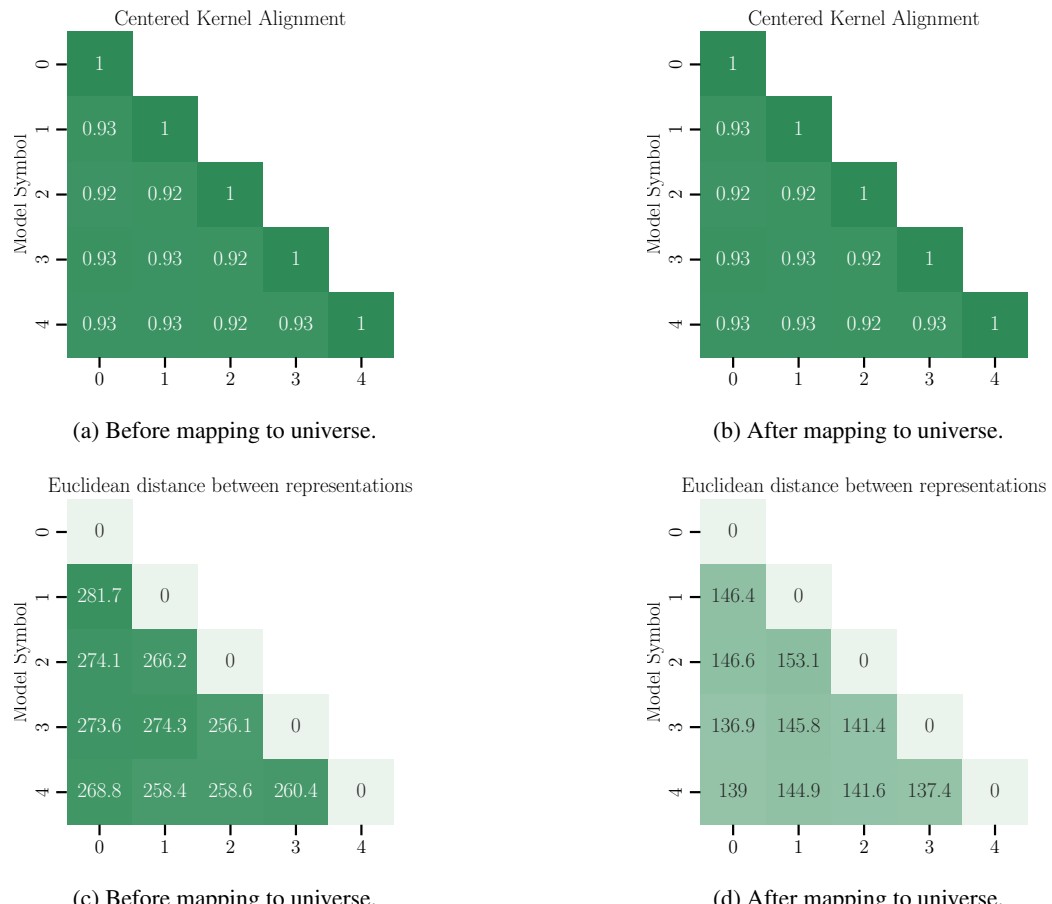

(a) Before mapping to universe.

(b) After mapping to universe.

(c) Before mapping to universe.

(d) After mapping to universe.

Figure 15: Cented Kernel Alignment and Euclidean distances of the representations of 5 `ResNet20` trained on `CIFAR10` with $2\times$ width.

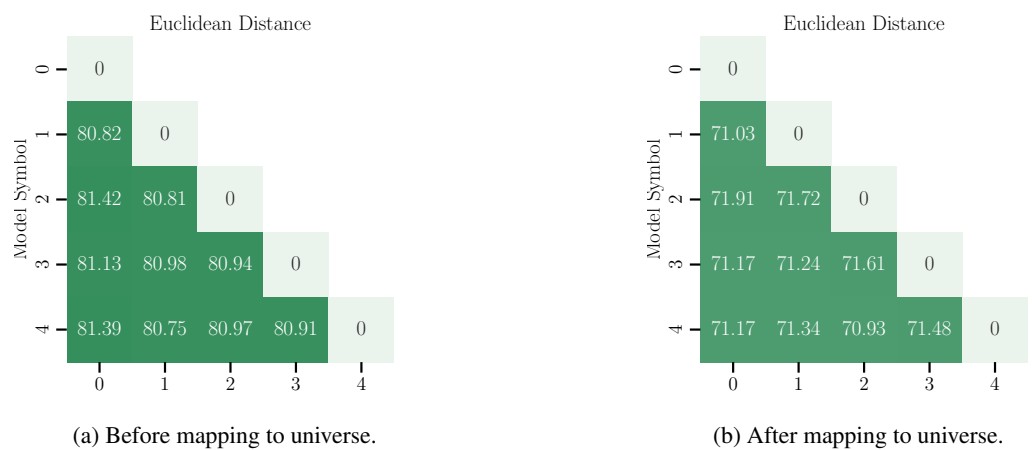

(a) Before mapping to universe.

(b) After mapping to universe.

Figure 16: Euclidean distance of the weights of 5 `ResNet20` trained on `CIFAR10` with $2\times$ width.

2. The particular composition of the match set has a significant impact over the matching and subsequent merge operation, yielding differences of up to $\approx 20$ accuracy points for the downstream model.

3. The standard deviations before the repair operation (red semi-transparent bars in the plots) are way lower when optimizing for the permutations over the superset of all 5 models; this suggests that the matching difficulty is spread over all the maps jointly, eventually yielding more stable results.

# D  Discussion

We discuss in this section the limitations of our work, as well as potential future societal impact.

## D.1  On the cycle-consistency of $C^2M^3$

Our method is natively cycle-consistent due to the mathematical formulation of the optimization problem. If we were to not desire cycle consistency, the matching method would fall back to the $n = 2$ Frank-Wolfe (FW) case presented in Section 3. One would then have to transform the pairwise matching problem to a $n$-way matching problem, *e.g.* by using the $n = 2$ FW procedure as matching step in the MergeMany [1] algorithm. Results for the $n = 2$ FW matching are reported in Table 3.

## D.2  Limitations

From what we have observed in our experiments, permutations satisfying linear mode connectivity of the models are hard to find for most architectures and datasets. In fact, given that there is no practical way to prove or disprove the conjecture for which most models end up in the same basin modulo permutations of the neurons, we cannot be sure that a certain set of models even allows finding such permutations, let alone that the permutations found are the optimal ones. We therefore encourage the community not to rely on the existence of such permutations in general. However, we have also shown that we can always find permutations that improve the resulting aggregated model, which is a promising practical result for model merging. As for all the existing works concerning linear mode connectivity and model merging, the resulting models that we obtain are sensible to a wide variety of factors, from training hyperparameters to the optimization algorithm used. Several works have already observed the phenomenon in practice: among these, Ainsworth et al. [1] mention among the known failure modes of their approaches models trained with SGD and too low learning rate, or ADAM coupled with too high learning rate. Jordan et al. [21] show that the chosen normalization layer incredibly affects the accuracy of the resulting merged model, while Qu and Horvath [30] observe learning rate, weight decay, and initialization method to play a strong role as well. Being a mostly empirical field, most of the technical choices that we make in our work mirror the ones made in previous works and are not based on a solid theoretical foundation. We therefore release all our code and encourage the community to investigate further on what training and optimization hyperparameters effect linear mode connectivity and model merging.

## D.3  Societal impact and broader vision

The work presented in this paper serves as an additional tool for the community to improve the efficiency of deep learning models. By merging models, we can reduce the computational cost of training and inference, as well as the memory footprint of the models. In fact, by aggregating the information of a set of models into a single one with the same architecture, practitioners can benefit of the effects of ensembling without incurring in its computational cost. Moreover, merging is in many cases a practical necessity to guarantee confidentiality and privacy of user data, as it allows to train models on different subsets of the data, *e.g.* originating from different clients, and then merge them to obtain a single model integrating all the information. This is particularly important in the context of federated learning, where the data is distributed among different clients and cannot be shared. We believe that the work presented in this paper can be a stepping stone towards more efficient and privacy-preserving deep learning models, and we encourage the community to further investigate the potential of model merging in these contexts.

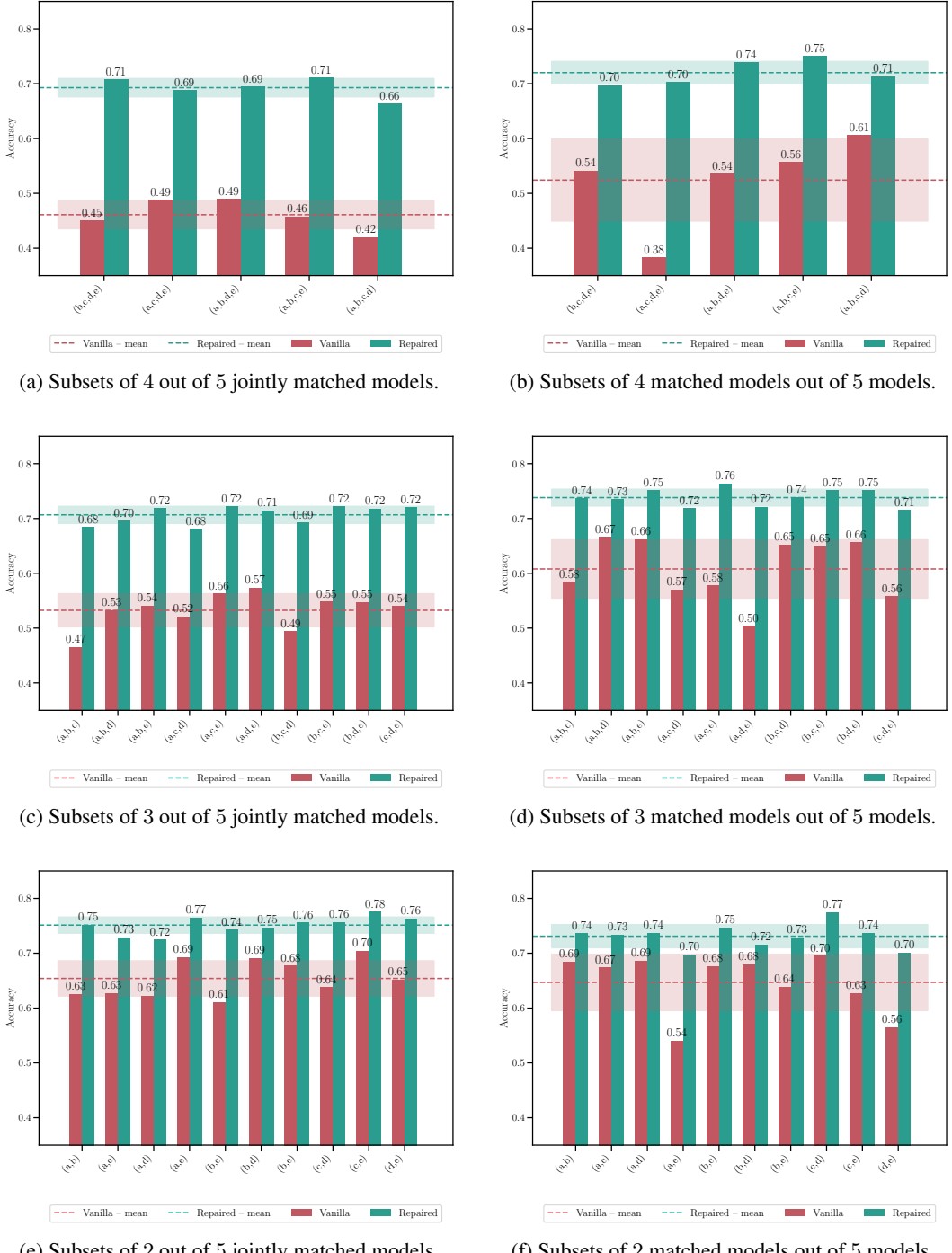

(a) Subsets of 4 out of 5 jointly matched models.

(b) Subsets of 4 matched models out of 5 models.

(c) Subsets of 3 out of 5 jointly matched models.

(d) Subsets of 3 matched models out of 5 models.

(e) Subsets of 2 out of 5 jointly matched models.

(f) Subsets of 2 matched models out of 5 models.

Figure 17: Accuracy of the resulting model when merging different model subsets. **(left)** performance of models obtained from aggregating subsets of $k < 5$ models that were matched jointly. **(right)** analoguous results for subsets of $k$ models that are instead matched independently, *i.e.*, by only optimizing for the permutations that align those $k$ models and discarding the remaining ones. The semi-transparent bands represent the standard deviation of the accuracy.

