# OpenReview forum: "$C^2M^3$: Cycle-Consistent Multi-Model Merging"
_NeurIPS.cc/2024/Conference — NeurIPS 2024 poster_

### Official Review · Reviewer_mu2n · 2024-07-10

**Soundness:** 3
**Presentation:** 3
**Contribution:** 3
**Rating:** 7
**Confidence:** 3

**Summary:**

The paper proposes a new, cycle-consistent method of merging more than two neural networks in weight space by simultaneously solving multiple permutation-based merge process. The key innovation is to ensure cycle consistency when merging $n>2$ models. The authors showed that the method, together with techniques like REPAIR, can lead to substantial benefits across various architectures and datasets.

**Strengths:**

1. The paper proposes a novel alignment algorithm that, by construction, ensures cycle consistency for merging multiple models. The authors specifically showed the benefit of merging models in a universe space $u$ as a bridge for model permutations. I personally found the idea of a universe space neat and sensible, and this method could be a nice advance in the field of model interpolation & merging.

2. Relatively extensive comparison with existing best methods and across different datasets and settings (including some analysis on the loss landscape, etc.)

3. The paper demonstrated compatibility of the method to adopt techniques like REPAIR which helps maintain good performance (i.e., practical enhancement over other prior methods)

**Weaknesses:**

1. While the concept of a universe weight space is nice, the presumption of neuron permutations and the related theoretical foundations are still based strongly on prior work.

2. I found the paper fail to analyze the pros and cons of the simultaneous global optimization in the main paper (though there are some in the appendix). Specifically, the Frank-Wolfe process will involve a $O(n^2L)$ loop which would be very costly compared to the more "local" approach by Ainsworth et al.

**Questions:**

1. The major concern I have is related to the global optimization process itself for model merging. How "easy" is this optimization process and why conditional GD? How does the optimization scale with wider/deeper networks? I suspect that the empirical observation in Fig. 12(a) in Appendix A.5 exactly suggests the instability in the optimization algorithm.

2. The authors reported the walk-clock time efficiency for merging 2 models in Appendix A.5. How slow is the method for merging $n>2$ models (e.g., say 5, 10), especially compared to setting one model as the "universe space" and perform $n-1$ pairwise optimizations?

3. How exactly is the merging factor $\alpha$ determined when trying to use REPAIR for multiple models in Table 1?

4. While by design the transformation to/from the universe weight space is invertible, but in practice depending on the quality of the optimization problem solving, the actual $P^{AC} \circ P^{CB} \circ P^{BA}$ could in theory still be not so close to the identity matrix. Maybe I missed something, so it'd be great if the authors can clarify.

---

> ### Author Rebuttal · Authors · 2024-08-07
>
> We thank the reviewer for the insightful comments and spot-on questions. We now proceed to address the raised concerns and answer the questions to the best of our capabilities.
>
> - **W1: model merging foundations based on prior work.** We agree with the reviewer that our work builds on top of existing work and partially leverages the same foundations. This is true, however, for each permutation-based model merging work, e.g. [1, 2, 3, 4, 5, 6], and we believe that sharing a set of common assumptions is inevitable in a subfield. We would also like to remark that **none of the existing works provably guarantees cycle consistency**, with the problem itself being overlooked. Therefore, while we acknowledge owing the existence of this research to prior work that has laid the foundations of the field, we argue that our work takes it a step further and, being theoretically grounded, has the potential to be itself a foundation for future research.
> - **W2: pros and cons of global vs local optimization.** We thank the reviewer for bringing this up. The answer is double-fold: i) **global optimization is required to enforce cycle consistency**, as it is an inherently global property; ii) **global optimization removes the arbitrariness in the layer iteration order**, which, as we report in Figure 4 and Table 6, results in a marked variance in the results.
> More in detail, it is not clear how cycle consistency, a global property of the model-level transformation, can be achieved using layer-local optimization. For this reason, we do not only choose global optimization due to the deterministic nature of its results but also because it is the only way to achieve our main and foremost objective — that of having cycle-consistent permutations between models. Agreeing with the reviewer about the importance of these considerations, we intend to use the extra page in the camera-ready version to clarify them.
> - **Q1: why conditional gradient descent.** We thank the reviewer for the question. Conditional gradient descent (Frank-Wolfe) is a particularly suitable algorithm for the problem we are trying to solve as it **only requires computing the gradient**, for which we derived a closed form, and enjoys two advisable properties: i) **monotonicity of the objective function** (see also Figure 11 in appendix) ii) **guaranteed convergence rates**, for which we added a proof in the appendix. Regarding the difficulty of the process, the monotonicity of the loss makes it arguably stable, even if the step size is decreasing in an alternating manner. As briefly explained in Appendix A. 5, we believe this behavior to reflect a fixed pattern in the optimization rather than an instability in the overall process. Regarding the scalability of the approach with larger networks, we report here its wall-clock time when merging n=2, 3 ResNet20 models having 1x, 2x, 4x, 8x, and 16x width, together with their number of parameters.
>
>
>     |  | 1x | 2x | 4x | 8x | 16x |
>     | --- | --- | --- | --- | --- | --- |
>     | # params | 292k | 1.166m | 4.655m | 18.600m | 74.360m |
>     | C2M3 time n=2 | 33.4s | 33.5s | 40.5s | 80.8s | 367.8s |
>     | C2M3 time n=3 | 32.9s | 83.18s | 91.0s | 162.0s | 715.8s |
>     | MergeMany time n=2 | 0.24s | 0.4s | 3.4s | 8.9s | 59.4s |
>     | MergeMany time n=3 | 1.2s | 4.1s | 19.5s | 105.8s | 892.3s |
>
>     As can be inferred from the table, the scaling laws depend on the complexity of the resulting matching problem and cannot be predicted merely from the number of parameters, with a 4-fold increase in parameters resulting in no increase in runtime for the first three columns, a double increase in the second-last column and a 5-fold increase in the last.   Compared to MergeMany, our approach enjoys a milder increase in running time when increasing the number of parameters. **We also included a rigorous proof determining the convergence rate of the algorithm in the rebuttal document**.
>
> - **Q2: Efficiency of the method for n>2.** We thank the reviewer for the question. To answer it, we computed the matching and merging time when merging $n=2, \dots, 10$ ResNet20 models with 4x width, as can be seen in Figure 1 of the rebuttal document. Compared to the pairwise baseline that maps all the models to a fixed one, our approach incurs a significantly steeper cost. However, Figure 2 (rebuttal document) shows that the latter also suffers from a performance decrease when increasing $n$ which is much more pronounced, making it not advisable for the task.
> - **Q3: Merging factor alpha.** We thank the reviewer for pointing out this missing detail. In all our experiments, we have used $\alpha = \frac{1}{n}$, where $n$ is the number of models to merge. This is indeed usually the hardest case, as it defines a point that is the furthest from the individual basins.
> - **Q4: cycle consistency.** We thank the reviewer for allowing us to clarify this aspect. Luckily, cycle consistency, i.e., the property for which applying a cycle of such transformations leads back to the starting point, holds in this case as long as each transformation is orthogonal. Looking at Figure 1 in the manuscript, it can be appreciated how mapping $A$ to another model $B$ means always passing through $U$  and then to $B$ with $P_B$, and that mapping $B$ to another model $C$ means passing through $U$ with $P_B^T$. This means that, as long as $P_B P_B^T = I$, the effect of mapping $A$ to $B$ is nullified. Since the proposed algorithm returns permutations (a subspace of orthogonal transformations), cycle consistency always holds for the resulting permutations.

---

> ### Author Response · Authors · 2024-08-07
>
> Thanking again the reviewer for their effort, we remain available for any further clarification.
>
> ### References
>
> [1] Ainsworth, Samuel, Jonathan Hayase, and Siddhartha Srinivasa. 2022. “Git Re-Basin: Merging Models modulo Permutation Symmetries.” In *The Eleventh International Conference on Learning Representations (ICLR), 2022*
>
> [2] Jordan, Keller, Hanie Sedghi, Olga Saukh, Rahim Entezari, and Behnam Neyshabur. 2023. “REPAIR: REnormalizing Permuted Activations for Interpolation Repair.” In *The Eleventh International Conference on Learning Representations (ICLR), 2023*
>
> [3] Navon, Aviv, Aviv Shamsian, Ethan Fetaya, Gal Chechik, Nadav Dym, and Haggai Maron. 2023. “Equivariant Deep Weight Space Alignment.” in *The Forty-first International Conference on Machine Learning (ICML), 2024*
>
> [4] Peña, Fidel A. Guerrero, Heitor Rapela Medeiros, Thomas Dubail, Masih Aminbeidokhti, Eric Granger, and Marco Pedersoli. “Re-Basin via Implicit Sinkhorn Differentiation.”, IEEE / CVF Computer Vision and Pattern Recognition Conference (CVPR), 2023
>
> [5] Singh, Sidak Pal, and Martin Jaggi. 2020. “Model Fusion via Optimal Transport.” In *Advances in Neural Information Processing Systems 33: Annual Conference on Neural Information Processing Systems 2020 (NeurIPS), 2020*
>
> [6] Horoi, Stefan, Albert Manuel Orozco Camacho, Eugene Belilovsky, and Guy Wolf. 2024. “Harmony in Diversity: Merging Neural Networks with Canonical Correlation Analysis.”, in *The Forty-first International Conference on Machine Learning (ICML), 2024*

---

### Official Review · Reviewer_TBfo · 2024-07-13

**Soundness:** 4
**Presentation:** 3
**Contribution:** 3
**Rating:** 6
**Confidence:** 4

**Summary:**

This paper further put forward a kind of cycle consistent Multi-Model Merging to merge models after permute simultaneously. It addresses the limitations in previous approaches that only handled pairwise merging. It uses a "universe" space to factorize permutations between models, optimizing all layer permutations simultaneously using the Frank-Wolfe algorithm.

**Strengths:**

- Deterministic result: independent of the random choice of layers
- The "universe" space to factorize permutations between models solves issues with previous methods that could accumulate errors when applying cyclic permutations.
- This paper has sufficient analyze and discussion on the model width, number of merged models, and linear mode connectivity in the universal basin.
- The algorithm to factorize is data-free (based on the Frank-Wolfe algorithm)

**Weaknesses:**

- Sensitive to hyper-parameters
- Lack of theoretical guarantee (I personally think there is not much researchers can do in this model-merging sub-domain. Especially in large networks)
- Exps are down on small datasets and small networks. Could do some exps on larger network and larger dataset
- What is the convergence speed given a different number of parameters? What about comparing to other algorithm such as MergeMany?

**Questions:**

- You mentioned: "the resulting models that we obtain are sensible to a wide variety of factors, from training hyperparameters to the optimization algorithm used" in the paper. Could you please elaberate more on this?
- What is your convergence speed comparing to git-rebasin (MergeMany). It would be great to see a Pareto Front showing that given a specific size of set of models, the trade-off between computation time and accuracy and see which algorithm occupies more Pareto Optimal solutions
- Is that possible to do some exps on CLIP-based/LLM (like the phi-3 which only has 3B parameters) larger networks? There are bunch of CLIP-fine-tuned networks with the same architecture and LLMs fine-tuned based on phi-3 etc. The exps are done only on EMNIST, CIFAR10 and CIFAR100. The largest network being used is ResNet-16/VGG-16 which are a bit out-of-data nowadays. (I know Git-rebasin was working on VGG-16 and ResNet 20 but still...)
- In figure 1.b, can you also calculate the cosine similarity? After all, L2 distance does not make much sense in high dimension space.
- The eq 3 is confusing, I am not sure if the author wants to sum p from 1 to n while not equals to q or sum (p,q) pairs from (1,1), (1,2), ..., (n-1,n) while p!=q. I am asking this because the author mentioned "In order to generalize to n models, we jointly consider all **pairwise**
problems". But the equation 3 does not show the "pairwise".
- What is the meaning of color in figure 3?

**Limitations:**

Yes, the author have mentioned the limitations well in the appendix.

---

> ### Author Rebuttal · Authors · 2024-08-07
>
> We thank the reviewer for the insightful comments and spot-on questions. We will now do our best to address each and every critique and question.
>
> **Sensitivity to hyperparameters**
>
> We thank the reviewer for raising this point. We would like, in fact, to clarify this aspect: **when, in the limitations, we refer to merging methods being sensible to hyperparameters, we do not refer to this particular approach**, but to the whole linear mode connectivity phenomenon. Different works have, in fact, observed this in practice: among these, Ainsworth et al. [1] when listing the known failure modes of their approaches, mention models trained with SGD and too low learning rate, or ADAM coupled with too high learning rate. Keller et al. [2] show that the chosen normalization layer incredibly affects the accuracy of the resulting merged model, while Qu et al. [3] observe learning rate, weight decay, and initialization method to play a strong role as well. We therefore argue that the sensitivity to hyperparameters is not a peculiarity of our approach, but something we observed throughout all the approaches we used and that has been confirmed in the existing literature. We added this clarification to the main manuscript.
>
> **Lack of theoretical guarantees**
>
> We thank the reviewer for the comment. As explicitly stated in the limitations of the paper, we share with the reviewer an overarching wish for an improved theoretical ground for model merging and linear mode connectivity. This is mostly due to the whole permutation-based model merging field building upon a conjecture that cannot be proven or disproven due to the huge number of possible neuron permutations. We would like, however, to remark that **this is not something that has to do with our research in particular but with the field in general** and that, differently from several works in the field, we provide a **principled approach that indeed holds some guarantees**. While this may be just a brick on a bumpy road, this brick is stable and allows further work to build upon it.
>
> **Small datasets and networks**
>
> We thank the reviewer for the suggestion.
>
> | Paper | Conference | Datasets | Architectures |
> | --- | --- | --- | --- |
> | Git Re-Basin [1] | ICLR22 | MNIST, CIFAR10, CIFAR100, ImageNet | MLP, VGG, ResNet |
> | REPAIR [2] | ICLR23 | MNIST, CIFAR10, CIFAR100, ImageNet | MLP, VGG, ResNet |
> | Deep Weight Space Alignment [3] | ICML24 | MNIST, CIFAR10, STL10  | MLP, VGG, CNN |
> | Re-Basin via Implicit Sinkhorn [4] | CVPR23 | ad hoc polynomial regression dataset, CIFAR10 | MLP, VGG |
> | Model fusion via optimal transport [5] | NeurIPS20 | MNIST, CIFAR10 | MLP, VGG, ResNet |
> | CCA-Merge [6] | ICML24 | CIFAR10, CIFAR100 | VGG, ResNet |
>
> As can be seen in the table, we are using the **most established set of architectures and datasets considered in all the previous and concurrent literature** in the field. This choice stems from two motivations: 1) for the **sake of comparison** and continuity with respect to previous works; 2) the **complexity of the architecture adds additional challenges** that must be taken into account and requires additional research that is not immediately relevant to the merging approach. In particular, the architecture suggested by the reviewer is a transformer-based architecture and requires ad hoc mechanisms to handle multi-head attention, positional embeddings, and the vast number of residual connections, in fact motivating stand-alone works to do this [4, 5].  We agree, however, with the reviewer that we are lacking a very large-scale case and, therefore, are also training ResNet50 endpoints on ImageNet. We aim to include these results as soon as they are available. Given our limited computational budget, these may be ready before the end of the discussion phase or for the camera-ready.
>
> **Convergence speed**
>
> We thank the reviewer for the question. We report here the wall-clock time when merging n=2,3 ResNet20 models having 1x, 2x, 4x, 8x, and 16x width, together with their number of parameters.
>
> |  | 1x | 2x | 4x | 8x | 16x |
> | --- | --- | --- | --- | --- | --- |
> | # params | 292k | 1.166m | 4.655m | 18.600m | 74.360m |
> | C2M3 time n=2 | 33.4s | 33.5s | 40.5s | 80.8s | 367.8s |
> | C2M3 time n=3 | 32.9s | 83.18s | 91.0s | 162.0s | 715.8s |
> | MergeMany time n=2 | 0.24s | 0.4s | 3.4s | 8.9s | 59.4s |
> | MergeMany time n=3 | 1.2s | 4.1s | 19.5s | 105.8s | 892.3s |
>
> As can be inferred from the table, the scaling laws depend on the complexity of the resulting matching problem and cannot be predicted merely from the number of parameters, with a 4-fold increase in parameters resulting in no increase in runtime for the first three columns, a double increase in the second-last column and a 5-fold increase in the last. Compared to MergeMany, our approach enjoys a milder increase in running time when increasing the number of parameters. **We also included a rigorous proof determining the convergence rate of the algorithm in the rebuttal document**.
>
> **Cosine similarity of the models**
>
> We thank the reviewer for the suggestion. We report here the cosine similarity between a model and the model obtained by cyclically applying the permutations obtained with git re-basin and C2M3 respectively.
>
> |  | (a, b, c, a) | (b, c, a, b) | (c, b, a, c) | (a, c, b, a)  |
> | --- | --- | --- | --- | --- |
> | git re-basin | 0.251 | 0.251 | 0.251 | 0.251 |
> | C^2M^3 | 1 | 1 | 1 | 1 |
>
> The result is analogous to that observed in the manuscript with the L2 distance, showing the lack of error accumulation in the proposed approach.
>
> **Ambiguity in equation 3**
>
> We thank the reviewer for pointing out this ambiguity. We confirm their understanding of the equation summing over all pairs of models `(1, 1), (1, 2), …, (n-1, n)` . For the sake of clarity, we replaced the equation in the manuscript with a triple sum, for each model `p=1, ..., n-1`, for each model `q=p+1, ..., n` and each layer `l=1, ..., L`.
>
> (TBC)

---

> ### Author Response · Authors · 2024-08-07
> **Rebuttal by Authors (2)**
>
> **Meaning of color in Figure 3**
>
> The color in Figure 3 represents the value of the loss in the loss landscape given by interpolations of the models. Red values indicate low-loss regions (basins), while blue values indicate high-loss regions. We added this information in the caption.
>
> Thanking again the reviewer for their effort, we remain available for any further clarification.
>
> **References**
>
> [1] Ainsworth, Samuel, Jonathan Hayase, and Siddhartha Srinivasa. 2022. “Git Re-Basin: Merging Models modulo Permutation Symmetries.” In *The Eleventh International Conference on Learning Representations (ICLR), 2022*
>
> [2] Jordan, Keller, Hanie Sedghi, Olga Saukh, Rahim Entezari, and Behnam Neyshabur. 2023. “REPAIR: REnormalizing Permuted Activations for Interpolation Repair.” In *The Eleventh International Conference on Learning Representations (ICLR), 2023*
>
> [3] Qu, Xingyu, and Samuel Horvath. "Rethink Model Re-Basin and the Linear Mode Connectivity." *arXiv preprint arXiv:2402.05966* (2024).
>
> [4] Imfeld, Moritz, et al. "Transformer fusion with optimal transport." In *The Twelfth International Conference on Learning Representations (ICLR), 2024*
>
> [5] Verma, Neha, and Maha Elbayad. "Merging text transformer models from different initializations." *arXiv preprint arXiv:2403.00986* (2024).

---

> > ### Comment · Reviewer_TBfo · 2024-08-11
> >
> > Thanks for addressing my questions. I am happy to increase the score.

---

> > > ### Author Response · Authors · 2024-08-12
> > >
> > > We are glad to hear that the questions have been addressed. We thank the reviewer for their effort and consideration.

---

### Official Review · Reviewer_dfjE · 2024-07-14

**Soundness:** 2
**Presentation:** 2
**Contribution:** 3
**Rating:** 5
**Confidence:** 2

**Summary:**

The paper introduces Cycle-Consistent Multi-Model Merging for merging neural networks by optimizing neuron permutations globally across all layers, ensuring cycle consistency when merging multiple models. Utilizing the Frank-Wolfe algorithm, this approach addresses inter-layer dependencies and guarantees that cyclic permutations result in the identity map. The method is generalized to handle more than two models by mapping each to a common universe space and thus enhancing alignment robustness.

**Strengths:**

- The paper introduces a method for merging neural networks by ensuring cycle consistency through global optimization of neuron permutations, addressing a limitation in existing pairwise approaches

- The use of the Frank-Wolfe algorithm for simultaneous layer optimization and incorporation of activation renormalization demonstrates an innovation and advantages through later experiments

**Weaknesses:**

- While the paper demonstrates the effectiveness of C2M3 across various (simple) architectures and (toy) datasets, it lacks a detailed analysis of the method's scalability to very large models or datasets. A more comprehensive evaluation of performance and computational requirements for larger-scale applications is desirable to understand its benefits

- Although the method shows promising results in experimental settings, the paper would be strengthened by including case studies or examples of real-world applications where C2M3 has been successfully implemented. This would help demonstrate the practical utility and robustness. Currently it is primarily focused on classification tasks.

- The paper lacks a deep theoretical analysis of the convergence properties and guarantees of the proposed method. Including theoretical insights or proofs regarding the convergence and stability of C2M3 would strengthen the work

- Other minor things:
    - In abstract the number of models is denoted by $N$, while $n$ is used in the main text
   - Fig 2 does not have "Figure 2" in the caption
   - the definition 1 is quite ambiguous, stating it as $A\approx B$. However, it is important to clarify the precise meaning of this approximation in a rigorous manner, especially when defining something clearly.

**Questions:**

See the weakness part above

**Limitations:**

Yes

---

> ### Author Rebuttal · Authors · 2024-08-07
>
> We thank the reviewer for their efforts in providing insightful comments and questions.
>
> **Method’s scalability to large models or datasets.**
>
> | Paper | Conference | Datasets | Architectures |
> | --- | --- | --- | --- |
> | Git Re-Basin [1] | ICLR22 | MNIST, CIFAR10, CIFAR100, ImageNet | MLP, VGG, ResNet |
> | REPAIR [2] | ICLR23 | MNIST, CIFAR10, CIFAR100, ImageNet | MLP, VGG, ResNet |
> | Deep Weight Space Alignment [3] | ICML24 | MNIST, CIFAR10, STL10  | MLP, VGG, CNN |
> | Re-Basin via Implicit Sinkhorn [4] | CVPR23 | ad hoc polynomial regression dataset, CIFAR10 | MLP, VGG |
> | Model fusion via optimal transport [5] | NeurIPS20 | MNIST, CIFAR10 | MLP, VGG, ResNet |
> | CCA-Merge [6] | ICML24 | CIFAR10, CIFAR100 | VGG, ResNet |
>
> As can be seen in the table, we are using the **most established set of architectures and datasets considered in all the previous and concurrent literature in the field**. This choice stems from two motivations: 1) for the **sake of comparison** and continuity with respect to previous works; 2) the **complexity of the architecture adds additional challenges** that must be taken into account and requires additional research that is not immediately relevant to the merging approach. In particular, transformer-based architectures require ad hoc mechanisms to handle multi-head attention, positional embeddings and the vast number of residual connections, in fact motivating stand-alone works to do this [8, 9].  We agree, however, with the reviewer that we are lacking a very large-scale case and, therefore, are also training ResNet50 endpoints on ImageNet. We aim to include these results as soon as they are available. Given our limited computational budget, these may be ready for the camera-ready.
>
> **Real-world applications**. We thank the reviewer for the suggestion. Tackling the problem of model merging, we inherit all the applicative domains suggested in relevant prior works, such as federated learning [1, 3], incremental learning [4], and continual learning [10]. While its foundational nature, in our opinion, makes examples of real-world applications out of scope for our paper, we share the curiosity of the reviewer in assessing its effectiveness in such a scenario. Therefore, we are currently setting up a federated learning experiment whose results we hope to share during the discussion phase.

---

> ### Author Response · Authors · 2024-08-07
>
> **Theoretical analysis: convergence properties and guarantees.**
>
> Following previous literature on the Frank-Wolfe algorithm [7], we know that FW obtains a stationary point at a rate of $\mathcal{O}(1 / \sqrt{t})$ on non-convex objectives with a Lipschitz-continuous gradient. We now prove that the considered objective function
>
> \begin{equation}
> \sum_{p=1}^{n-1} \sum_{q=p+1}^{n}  \sum_{\ell=1}^L \langle (P_{\ell}^p )^\top W_\ell^p P_{\ell -1}^p, (P_{\ell}^q)^\top W_{\ell}^q P^q_{\ell -1} \rangle
> \end{equation}
>
> Has a Lipschitz-continuous gradient. We recall that, for each layer permutation $P^A = \{P_1^A, P_2^A, \ldots, P_{L}^A\}$ of model $A$, we can define the gradient of our objective function relatively to the model $B$ we are matching towards:
>
> \begin{equation}
> f(P_l^A) = \nabla^{\text{rows},P_l^A} + \nabla^{\text{cols},P_l^A} + \nabla^{\text{rows},\leftrightarrows,P_l^A} + \nabla^{\text{cols},\leftrightarrows,P_l^A} = \left[W^A_l P_{l-1}^A (P^B_{l-1})^\top (W^B_{l})^\top + (W^A_{l+1})^\top P_{l+1}^A (P^B_{l+1})^\top W^B_{l+1}\right] P^B_{l} + \left[W^B_{l} P_{l-1}^B(P^A_{l-1})^\top (W^A_{l})^\top + (W^B_{l+1})^\top P_{l+1}^B (P^A_{l+1})^\top W^A_{l+1}\right] P^A_{l}
> \end{equation}
>
> To prove Lipschitz continuity, we need to show there
> exists a constant $C$ such that
> $\forall p\in[1,n] l\in[1,L] \lVert f(P_\ell^p) - f(Q_\ell^p) \rVert \leq C \lVert P_\ell^p - Q_\ell^p \rVert$.
> To simplify passages, we only consider a fixed $l$ and perform a
> generic analysis. We begin by observing that
> $$
> f(P_l^p) - f(Q_l^p) =
> \sum_{q\in[1,n]\setminus \{p\}}
> \left[ W^p_{l} P_{l-1}^p (P^q_{l-1})^\top (W^q_{l})^\top \right. + \left. (W^p_{l+1})^\top P_{l+1}^p (P^q_{l+1})^\top W^q_{l+1}\right](P^q_{l} - Q^q_{\ell}) +
> \left[ W^q_{l} P_{l-1}^q(P^p_{l-1})^\top (W^p_{l})^\top \right. + \left. (W^q_{l+1})^\top P_{l+1}^q (P^p_{l+1})^\top W^p_{l+1}\right]  (P^p_{l}-Q^p_{l})
> $$
> The last form of the above equation can be rewritten as
> a sum of the two sums:
> $$
> \sum_{q\in[1,n]\setminus \{p\}}  \left[W^p_{l} P_{l-1}^p (P^q_{l-1})^\top (W^q_{l})^\top \right. +  \left.(W^p_{l+1})^\top P_{l+1}^p(P^q_{l+1})^\top W^q_{l+1}\right] (P^q_{l} - Q^q_{l})  +
> \sum_{q\in[1,n]\setminus \{p\}}  \left[W^q_{l} P_{l-1}^q(P^p_{l-1})^\top (W^p_{l})^\top \right. + \left.(W^q_{l+1})^\top P_{l+1}^q (P^p_{l+1})^\top W^p_{l+1}\right] (P^p_{l}-Q^p_{l})
> $$
> Since the first term does not depend on either
> $P_l^p$ or $Q_l^p$, we assume as a worst case that its norm is 0.
> Then, we remove transposes (since $\lVert M \rVert = \lVert M^\top \rVert$) and apply the triangle
> inequality and the sub-multiplicative property of matrix norms:
> $$
>  \lVert f(P_l^p) - f(Q_l^p) \rVert \leq
> \sum_{q\in[1,n]\setminus \{p\}} \lVert P^p_l-Q^p_{l}\rVert   \left( \lVert W^q_l\rVert   \lVert P_{l-1}^q\rVert  \lVert P^p_{l-1}\rVert   \lVert W^p_l\rVert   \right. + \left. \lVert W^q_{l+1}\rVert   \lVert P_{l+1}^q\rVert   \lVert P^p_{l+1}\rVert    \lVert W^p_{l+1}\rVert  \right)
> $$
> Let
> $$
> C = \max_{q\in[1,n]\setminus \{p\}} ( \{ \lVert W^q_l \rVert \lVert P_{l-1}^q\rVert \lVert P_{l-1}^p\rVert \lVert W^p_l\rVert + \lVert W^q_{l+1}\rVert \lVert P_{l+1}^q\rVert \lVert P^p_{l+1}\rVert \lVert W^p_{l+1}\rVert } )
> $$.
> Then,
> $$
> \lVert f(P_l^p) - f(Q_l^p) \rVert \leq C  \sum_{q\in [1,n]\setminus p} \lVert P_l^p - Q_l^p \rVert = C (n-1) \lVert P_l^p - Q_l^p \rVert
> $$
>
> we conclude that $f(P_l^p)$ is Lipschitz continuous for all models
> and all layers, with Lipschitz constant $C(n-1)$ depending on both the
> norm of the weights matrices and the number of models.

---

> ### Author Response · Authors · 2024-08-07
>
> **Inconsistencies:**
>
> We thank the reviewer for accurately spotting the naming inconsistency, we have promptly fixed the issue in the manuscript by making $n$ lowercase everywhere. Analogously, we added a figure-level caption to Figure 2 which incorrectly had only two subfigure-level captions. The caption states “*Existing methods accumulate error when cyclically mapping a model through a series of permutations, while $C^2M^3$ correctly maps the model back to the starting point.*”.
>
> Regarding definition 2.1, we used the definition as presented in Git Re-Basin [1]. We believe, however, the reviewer’s comment to be spot on, as the current formulation may hinder clarity. We therefore replaced the $\mathcal{L}(\Theta_A) \approx \mathcal{L}(\Theta_B)$ assumption by instead asking that the two points correspond to weights of neural networks trained to convergence with SGD.
>
> **References**
>
> [1] Ainsworth, Samuel, Jonathan Hayase, and Siddhartha Srinivasa. 2022. “Git Re-Basin: Merging Models modulo Permutation Symmetries.” In *The Eleventh International Conference on Learning Representations (ICLR), 2022*
>
> [2] Jordan, Keller, Hanie Sedghi, Olga Saukh, Rahim Entezari, and Behnam Neyshabur. 2023. “REPAIR: REnormalizing Permuted Activations for Interpolation Repair.” In *The Eleventh International Conference on Learning Representations (ICLR), 2023*
>
> [3] Navon, Aviv, Aviv Shamsian, Ethan Fetaya, Gal Chechik, Nadav Dym, and Haggai Maron. 2023. “Equivariant Deep Weight Space Alignment.” in *The Forty-first International Conference on Machine Learning (ICML), 2024*
>
> [4] Peña, Fidel A. Guerrero, Heitor Rapela Medeiros, Thomas Dubail, Masih Aminbeidokhti, Eric Granger, and Marco Pedersoli. “Re-Basin via Implicit Sinkhorn Differentiation.”, IEEE / CVF Computer Vision and Pattern Recognition Conference (CVPR), 2023
>
> [5] Singh, Sidak Pal, and Martin Jaggi. 2020. “Model Fusion via Optimal Transport.” In *Advances in Neural Information Processing Systems 33: Annual Conference on Neural Information Processing Systems 2020 (NeurIPS), 2020*
>
> [6] Horoi, Stefan, Albert Manuel Orozco Camacho, Eugene Belilovsky, and Guy Wolf. 2024. “Harmony in Diversity: Merging Neural Networks with Canonical Correlation Analysis.”, in *The Forty-first International Conference on Machine Learning (ICML), 2024*
>
> [7] Lacoste-Julien, S. (2016). Convergence Rate of Frank-Wolfe for Non-Convex Objectives. *ArXiv, abs/1607.00345*.
>
> [8] Imfeld, Moritz, et al. "Transformer fusion with optimal transport." In *The Twelfth International Conference on Learning Representations (ICLR), 2024*
>
> [9] Verma, Neha, and Maha Elbayad. "Merging text transformer models from different initializations." *arXiv preprint arXiv:2403.00986* (2024).
>
> [10] Marczak, Daniel, et al. "MagMax: Leveraging Model Merging for Seamless Continual Learning." ECCV 2024

---

> > ### Author Response · Authors · 2024-08-12
> > **Additional results**
> >
> > As anticipated in the rebuttal, we have run our framework in a federated learning scenario. To this end, we have used the state-of-the-art federated learning library Flower and employed our matching scheme over a set of 10 clients over CIFAR10, each adopting a small CNN model. We observed the following:
> > 1. When all the clients start from the same initialization, our approach has no benefit and falls back to standard averaging. In fact, the optimization process quickly returns identity matrices as permutations, suggesting the models already share the same basin.
> > 2. When instead we initialize the clients from different random initializations, our approach visibly outperforms FedAVG.
> >
> > We report here the accuracy for 50 aggregation rounds, with each client training for 20 local epochs. We report the results every 5 rounds for brevity.
> > | | 1 | 5 | 10 | 15 | 20| 25| 30| 35| 40| 45|
> > | -- | --  | -- | -- | -- | -- | -- | -- | -- | -- | --|
> > | FedAvg |0.0942 | 0.394 | 0.4972 | 0.5517 | 0.5699 | 0.5893 | 0.6018 |0.6063 | 0.6099 | 0.6136|
> > | C2M3 | 0.0941 | **0.4234** | **0.5193** | **0.5555** | **0.5783** | **0.5978** | **0.6077** | **0.6165** | **0.618** | **0.622** |
> >
> > If we increase the number of local epochs, the benefits get more pronounced. This is in line with the intuition that standard averaging becomes less effective when clients drift due to prolonged local training and too infrequent aggregation.
> >
> > | | 1 |2| 3| 4| 5| 6| 7| 8| 9| 10 |
> > | -- | --  | -- | -- | -- | -- | -- | -- | -- | -- | --|
> > | FedAvg |0.0942 | 0.2638 | 0.3543 | 0.3825 | 0.4165 | 0.4505 | 0.4742 | 0.4994 | 0.5169 | 0.5317|
> > | C2M3 | **0.0947** | **0.3303** | **0.3899** | **0.4441** | **0.4764** | **0.4968** | **0.5184** | **0.5334** | **0.5434** | **0.5536**|
> >
> > While these results are not sufficient to claim an overall supremacy of the approach for the task due to the limited evaluation and choice of models, they show the approach to be promising for the problem and encourage further research.
> >
> > Thanking the Reviewer for their suggestion, we kindly encourage them to revise their score if they consider their concerns to be addressed; otherwise, we remain available for any further clarification.

---

### Author Rebuttal · Authors · 2024-08-07

We thank all the reviewers for their insightful comments and questions. We are happy to see the contribution of our work be appreciated, with reviewers finding the work innovative, empirically advantageous, and addressing a limitation in existing approaches (djfE). The concept of a universe space was found to be neat and sensible, and a nice advancement to the field (mu2n). Moreover, we are happy to see the benefits of the approach acknowledged, being deterministic, data-free and avoiding the accumulation of error when permuting cyclically (TBfo). Finally, we are glad to see that the experiments and comparisons with the best existing methods were found to be extensive (mu2n) and the analysis and discussions to be sufficient (TBfo).
Having done our best to address all the raised concerns, we remain available for any further clarification or doubt.

---

### Decision · Program_Chairs · 2024-09-25

**Decision:**

Accept (poster)

**Comment:**

The paper identifies a limitation in the existing model merging methods and proposes adding a constraint of cycle consistency through their optimization formulation. The merging of the models is done through global optimization and the authors propose a Frank-Wolfe algorithm for the same. The solution is further improved through layer-renormalization. All the reviews agree on benefits of the cycle consistency constraint for model merging as well as on the novelty of proposed Frank-Wolfe algorithm. The strengths of this paper lies in the novel constraint, the proposed algorithm and its benefit for model fusion.

The main concerns that have been raised in the reviews are on

1. Experimentation only done with smaller datasets & networks,
2. Convergence properties of the proposed algorithms,
3. Real world case-study which demonstrates practical use of the model merging methods.

The rebuttal from the authors address some of these concerns -
1. The authors claim that they are using the same datasets as Git-rebasin (prior-works) to benchmark, as well as (currently) training ResNet50 endpoints on ImageNet;
2. For convergence they have included additional proofs as well as shared a table clocking the model merging;
3. For the real world case study they share a Fed-learning example which seems a simpler example.

I recommend *acceptance of this paper with minor revisions*. Following are the specifics that should be addressed in the final version -
1. Please include additional results using ResNet50 and ImageNet.
2. Please include convergence speed tables as well as proofs at appropriate places.
3. Please include discussions on results for Fed-learning highlighting the cases when the method works and the cases
4. A discussion or ablation experiment which considers only the FW algorithm without the consistency constraint.